EMBO
Molecular Medicine

# Reprogramming-derived gene cocktail increases cardiomyocyte proliferation for heart regeneration

Yuan-Yuan Cheng[1,2], Yu-Ting Yan[1,2], David J Lundy[2], Annie HA Lo[2], Yu-Ping Wang[2], Shu-Chian Ruan[2], Po-Ju Lin[2] & Patrick CH Hsieh[1,2,3,*]

## Abstract

Although remnant cardiomyocytes (CMs) possess a certain degree of proliferative ability, efficiency is too low for cardiac regeneration after injury. In this study, we identified a distinct stage within the initiation phase of CM reprogramming before the MET process, and microarray analysis revealed the strong up-regulation of several mitosis-related genes at this stage of reprogramming. Several candidate genes were selected and tested for their ability to induce CM proliferation. Delivering a cocktail of three genes, FoxM1, Id1, and Jnk3-shRNA (FIJs), induced CMs to re-enter the cell cycle and complete mitosis and cytokinesis *in vitro*. More importantly, this gene cocktail increased CM proliferation *in vivo* and significantly improved cardiac function and reduced fibrosis after myocardial infarction. Collectively, our findings present a cocktail FIJs that may be useful in cardiac regeneration and also provide a practical strategy for probing reprogramming assays for regeneration of other tissues.

**Keywords** cardiomyocyte proliferation; gene therapy; heart regeneration; myocardial infarction; reprogramming
**Subject Categories** Cardiovascular System; Regenerative Medicine

## Introduction

Cardiovascular diseases are a leading cause of death in developed countries (Khurana *et al*, 2005). Heart failure remains a worldwide public health problem due to poor heart regeneration after injury. However, the strong regenerative ability observed in lower vertebrates by remnant cardiomyocyte (CM) proliferation after injury provides hope for the development of effective treatments (Poss *et al*, 2002; Olson, 2006; Jopling *et al*, 2010; Gupta & Poss, 2012). Cardiac repair in mammals does not demonstrate proliferative ability equivalent to that in lower vertebrates, implying that there is some mechanism that has altered the regenerative

abilities in these phylogenetically close groups. In mammals, CMs undergo a final round of DNA synthesis and karyokinesis, become binucleated, and subsequently exit the cell cycle during the first week after birth (Li *et al*, 1997a,b). Several genetic alterations have recently been reported that support the re-entry of CMs into the cell cycle for proliferation (Ball & Levine, 2005; Engel *et al*, 2006; Mahmoud *et al*, 2013; Sengupta *et al*, 2013), yet the efficacy is not high enough to have great therapeutic potential. Genetic alterations which are able to improve CM proliferation effectively enough for heart regeneration after injury remain unclear.

Somatic cells can be reprogrammed into induced pluripotent stem cells (iPSCs) that can differentiate into multiple lineages and proliferate at a high rate for maintenance of pluripotency (Takahashi & Yamanaka, 2006). Many studies have shown the therapeutic potential of iPSCs in heart regeneration in different species (Chong *et al*, 2014; Masumoto *et al*, 2014). However, sophisticated CM differentiation processes, low reprogramming efficiency, and concerns over tumorigenicity have led to clinical hurdles to the application of iPSCs in cardiac regeneration (Wamstad *et al*, 2012; Zhang *et al*, 2013). A high rate of proliferation is essential for successful iPSC generation (Ruiz *et al*, 2011), suggesting that the gain of proliferative capacity might be a key for CM reprogramming. Therefore, we theorized that understanding the process of CM reprogramming may yield clues into the way by which CMs re-enter the cell cycle. These clues may then be exploited in order to promote efficient heart regeneration after injury. Unfortunately, the exact process by which CMs initiate and complete this reprogramming procedure remains unknown.

Recent studies have provided insight into this sophisticated reprogramming process. Several different markers are used to define each stage of mouse embryonic fibroblast (MEF) reprogramming including Thy1[+], Thy1[−], SSEA-1[+], and SSEA-1[+]/Oct4[+] (Hansson *et al*, 2012; Polo *et al*, 2012). Interestingly, SSEA-1 expression is confirmed as an intermediate cell marker leading to the commitment of iPSC generation, and the Thy1-negative stage is reported as necessary for proliferation enhancement, metabolic change, and loss of identity for initiating reprogramming (Hansson *et al*, 2012; Polo *et al*, 2012). These findings

1 Graduate Institute of Life Sciences, National Defense Medical Center, Taipei, Taiwan
2 Institute of Biomedical Sciences, Academia Sinica, Taipei, Taiwan
3 Department of Surgery, Institute of Medical Genomics and Proteomics, Institute of Clinical Medicine, National Taiwan University & Hospital, Taipei, Taiwan
*Corresponding author. Tel: +886 2 27899170; Fax: +886 2 27858594; E-mail: phsieh@ibms.sinica.edu.tw

imply that the gain of proliferative capacity takes place at an early stage in the reprogramming process, before SSEA-1 expression. Although the reprogramming of CMs into iPSCs has been reported (Rizzi *et al*, 2012; Xu *et al*, 2012), it is still unclear whether CMs, which have poor proliferative capacity, undergo similar reprogramming processes to MEFs, which have a high proliferation rate. Moreover, it is challenging to pinpoint the specific phase at which CMs return to the complete cell cycle during the reprogramming process. We hypothesized that the key factors activated at defined proliferative time points during early CM reprogramming may provide important information for cardiac regeneration. Therefore, we sought to closely examine the changes occurring during early CM reprogramming by microarray analysis, and to identify the ideal genetic alteration which could yield the greatest therapeutic potential after MI.

## Results

### Cardiomyocytes regain proliferative capabilities during early reprogramming

A system using doxycycline-regulated OSKM transgenic mice was used to execute reprogramming (Carey *et al*, 2011). In order to isolate a high-purity population of CMs from the murine neonatal heart, we took advantage of a technique utilizing the low-toxicity mitochondrial dye tetramethylrhodamine methyl ester perchlorate (TMRM) to select for CMs in fluorescence-activated cell sorting (FACS). CMs represent the highest staining population in the heart (Hattori *et al*, 2010). Furthermore, in order to obtain the highest purity sample of CMs possible, CM and non-CM (NC) populations from the same hearts were separated following a double-gated criterion based on mitochondrial dye TMRM as well as cell size and mass (Fig 1A). Every cell in the TMRM$^{++}$ population showed cardiac troponin I (cTnI) and α-sarcomeric actinin (α-SA) expression and clear sarcomere structure with rhythmic beating after attaching in dishes for 6 days, firmly identifying them as CMs (Fig 1B). The TMRM$^{+}$ population of NCs showed vimentin expression and so most were identified as cardiac fibroblasts (Fig 1B). Clear colony formation (AP$^{+}$ colonies) was detected 6 days after doxycycline treatment, and SSEA-1 expression was confirmed in both CMs and NCs. In order to further confirm that derived SSEA-1$^{+}$ colonies did indeed originate from CMs, triple transgenic mice OSKM/myh6-mER-Cre-mER/ZEG were generated and only isolated CMs were labeled green following 4-OH tamoxifen treatment *in vitro* (Fig 1C). These green-labeled CMs were then reprogrammed and the derived SSEA-1$^{+}$ colonies were formed in green after doxycycline treatment for 6 days,

showing that these colonies did indeed originate from isolated CMs (Fig 1D and E). Suitable time points to represent each part of the initial stage of the reprogramming process were determined to be days 0, 2, 4, and 6 based on the morphological changes (Fig 1F). Gene expression profiles at each time point were then assessed by microarray analysis.

Gene profiling at each time point during early reprogramming of CMs and NCs is shown in Fig 2A. We examined expression levels of over 15,000 genes during the time period measured. The expression profiles were divided into eight clusters based on the different expression patterns observed during early reprogramming in both cell types (Fig 2A and B). Gene ontological analysis was used to compare the changes in gene expression which occurred at each time point for each cell population, using the previous time point as a comparative baseline. The results show clear differences between CM and NC populations as they progress through the reprogramming process (Fig 2C). The most significant changes in the CM population are the increased expression of genes associated with mitosis (Cluster 1) and the down-regulation of genes associated with chemokine production (Cluster 6) on D2, whereas the NC population undergoes mesenchymal–epithelial transition (MET) on D2, showing increased expression of epidermal differentiation markers (Cluster 3) and a large increase in expression of genes associated with sterol biosynthesis (Cluster 2). On the other hand, the genes related to MET process were highly measured until D4 of CM reprogramming. This process continued through D4 and D6 in CMs, showing up-regulation of genes associated with tight junction formation (Cluster 4). These changes were observed much earlier (D2–D4) in the NC population. Similarly, up-regulation of stem cell maintenance markers (Cluster 5) began in NCs on D4 but did not begin until D6 of CM reprogramming. Mitochondrial or cardiac-related genes were specifically down-regulated at CM-D4 or CM-D6 (Cluster 7 and Cluster 8). Some genes in specific functional groups were also confirmed by real-time PCR and showed identical trends as the microarray results (Appendix Fig S1). Thus, it appears that CMs follow a time-delayed sequence of reprogramming events, having to first regain proliferative ability during the first 2 days of the process, before progressing through reprogramming in the same manner as NCs.

In order to better analyze this time-delayed sequence of events, changes in gene expression between the NC and CM populations were correlated. Comparative analysis by the Pearson correlation coefficient (Fig 2D) showed that the gene expression profile at CM-D4 is similar to NC-D2 or NC-D4, and CM-D6 is more similar to NC-D4 than NC-D6, again demonstrating the time delay phenomenon. Further analysis by principle component analysis (Fig 2E) showed that CMs require more "effort" to get to the same reprogramming

---

**Figure 1. Early reprogramming process of CMs and NCs.**

A   Flow cytometry of cells isolated from heart tissue when scaled against TMRM staining and also FSC and SSC parameters of flow cytometry.

B   Cardiac troponin I (cTnI) and α-sarcomeric actinin (α-SA) staining of the TMRM$^{++}$ population (CMs) and vimentin staining of the TMRM$^{+}$ population (NCs). Scale bar represents 100 μm.

C   Schematic showing early reprogramming processes of CMs and NCs isolated from triple transgenic mice.

D   EGFP expression in iPSC-like colonies derived from CMs or NCs isolated from triple transgenic mice. Scale bar represents 100 μm.

E   EGFP expression combined with immunofluorescence staining of SSEA-1 and DAPI of iPSC-like colonies derived from CMs isolated from triple transgenic mice. Scale bar represents 100 μm.

F   Morphological changes of isolated CMs and NCs at different time points before AP-positive colony formation at day 6. Scale bar represents 100 μm.

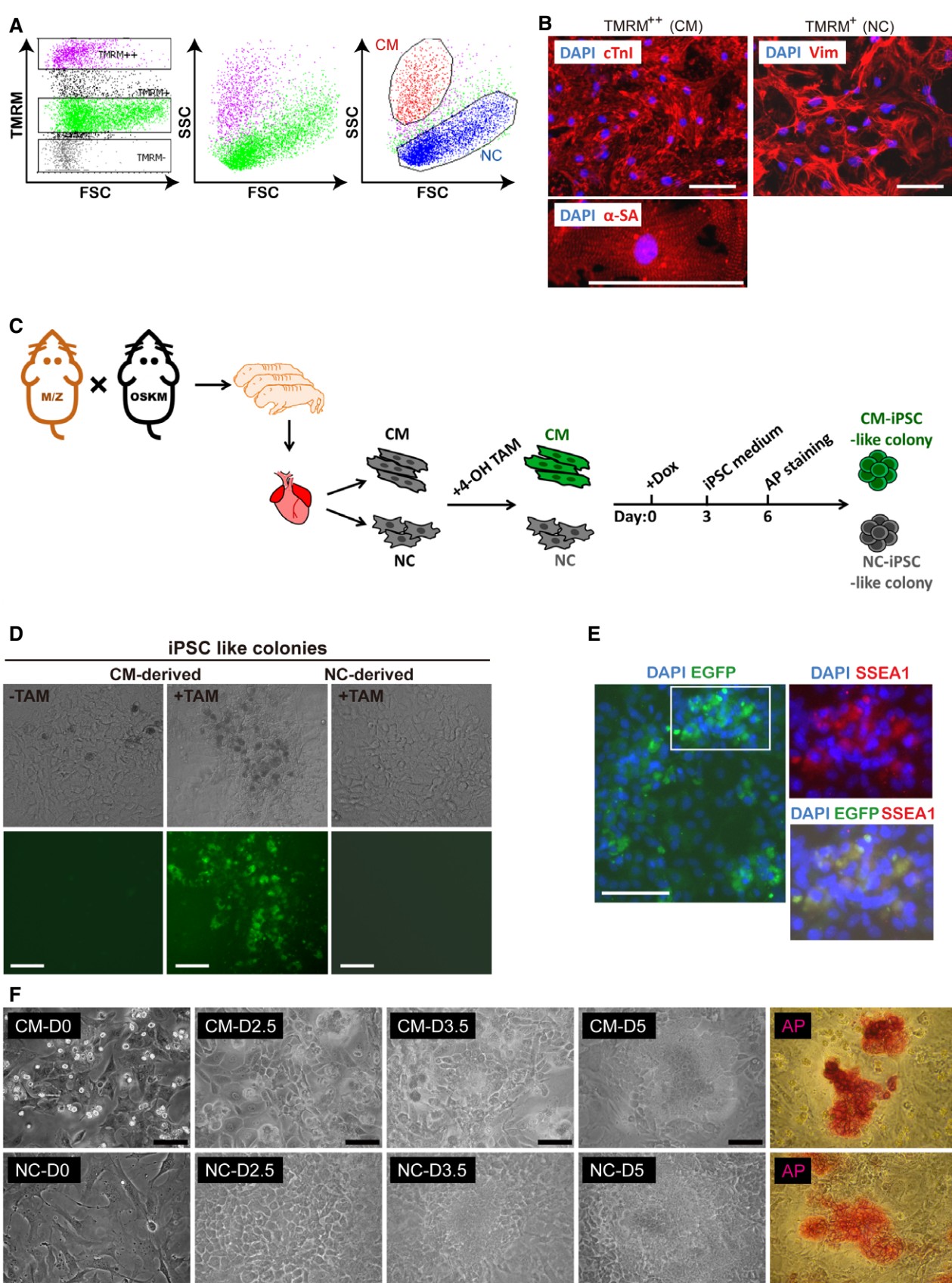

**Figure 1.**

**Figure 2. Microarray analysis of early CM reprogramming process compared to NCs.**

A   Gene profiling of CM and NC at D0–D6 as shown clustered into eight categories. Red indicates up-regulated genes while green indicates down-regulated genes ($n = 3$, $n = 2$ for NC-D6).

B   Expression patterns of all genes per cluster by gray lines. Black line shows the mean value.

C   Gene ontological analysis of early reprogramming at different time points.

D   Correlative analysis representing the relationships between CM and NC reprogramming specifically at days 2, 4 and 6. Red indicates closely related groups, while green indicates significantly different groups.

E   Principle component analysis (PCA) analysis indicating the routes during early reprogramming of CMs and NCs.

F   Schematic showing early reprogramming processes of CMs and NCs.

stage as NCs on D2. However, once the CM population reaches the same stage of reprogramming, the two populations follow very similar reprogramming processes. Over the same time period, CMs had only completed half of the initiation process in comparison with NCs. A schematic diagram outlining this process is presented in Fig 2F.

## Determination of the gene cocktail with the potential of increasing cardiomyocyte proliferation *in vitro* and *in vivo*

Microarray analysis showed strong up-regulation of genes associated with mitosis during the second day of CM reprogramming. Therefore, individual genes which showed at least a twofold up-regulation or down-regulation were singled out for further analysis. This included 488 up-regulated and 47 down-regulated genes (Appendix Table S1). Nine genes mostly belonging to the mitosis functional group were selected, and their D2:D0 ratio of RNA expression in CM and NC populations was compared. As shown in Fig 3A, FoxM1, Id1, Hmgb2, and Jnk3 showed the largest changes in expression, especially in CMs. Following the experimental procedure presenting in Fig 3B, statistical analysis showed that only FoxM1 (F) and Id1 (I) overexpression and Jnk3 inhibitor (Ji) treatment were able to increase CM proliferation as measured by the $\alpha$-MHC$^+$/Ki67$^+$ population (Fig 3C). Combining all three into one treatment (FIJi) further increased proliferation sevenfold compared to the control (Fig 3C). In order to more specifically knockdown Jnk3 expression, two Jnk3-shRNAs with more than 90% knockdown efficiency were tested (Appendix Table S2). However, only one specific adeno-FIJs could efficiently enhance CM proliferation to the same level as the inhibitor-treated CMs (Fig 3D). In addition, histone H3 phosphorylation (H3P) was used to confirm that CMs were carrying out the mitosis process. As shown in Fig 3E, FIJs-treated CMs showed a sevenfold increase in the H3P$^+$ population in comparison with control-treated cells. Aurora kinase B (AURKB) staining was used to confirm commitment to cytokinesis. During anaphase, AURKB was localized in the middle of the two sets of chromosomes; and AURKB showed as two dots in the middle of two separating daughter cells during telophase (Fig 3E). FIJs-treated CMs showed a sixfold increase in AURKB$^+$ population in comparison with control-treated cells. In terms of morphology, these CMs showed lower $\alpha$-MHC expression and less organized sarcomeres in Fig 3E, as is typical of proliferating CMs, as mentioned previously (Eulalio *et al*, 2012; Naqvi *et al*, 2014). Furthermore, time-lapse video showed that FIJs-treated CMs complete the whole cell cycle to produce two daughter cells, and total cell numbers were calculated to show more than 5-times increase in FIJs-treated CMs than control group (Appendix Fig S2A and B). These results demonstrated that CMs underwent mitosis with complete cytokinesis following FIJs treatment. Furthermore, the FIJs treatment showed no supportive of CF proliferation due to endogenous higher expression of FI and lower expression of Jnk3 in CFs compared to CMs (Appendix Fig S3), suggesting that the FIJs-induced proliferation is specific to CMs.

Changes in the expression of Oct4, FoxM1, Id1, and Jnk3 were further confirmed *in vivo* by injecting doxycycline into OSKM transgenic mice and isolating adult CMs after 2 days (Fig 3F). Overexpression of Oct4 was confirmed for the successful treatment of doxycycline injection *in vivo* (Fig 3G). As expected, FoxM1 and Id1

showed higher expression and Jnk3 was significantly down-regulated compared to control CMs (Fig 3G). In addition, significantly higher H3P$^+$ population of adult CMs (2x) was found 2 days after doxycycline treatment (Fig 3H).

## A defined gene cocktail significantly increases cardiomyocyte proliferation in both neonatal and adult mice

The transcriptional expression levels of FoxM1, Id1, and Jnk3 were examined by collecting whole hearts at different stages throughout development (Appendix Fig S4). The expression pattern of down-regulated FoxM1 and Id1 with up-regulated Jnk3 in the heart after birth may be related to the loss of the proliferative ability in CMs. Furthermore, FIJs treatment was administrated to the thoracic cavity at the left parasternal position of P1 neonatal mice as described in Fig 4A. After 12 days, the heart size of FIJs-treated mice was visibly larger and the heart-to-body weight ratio was also significantly higher than control-treated mice (Fig 4B). Critically, this enlarged size can be attributed to increased proliferation rather than hypertrophy since the H3P$^+$ population in FIJs-treated mice was three times higher than control mice (Fig 4C), and interestingly, we also noted that the majority of proliferating CMs were mononucleated rather than multinucleated.

Furthermore, the ability of FIJs treatment to enhance adult CM proliferation *in vivo* was demonstrated by direct injection into the heart of 10-week-old mice (Fig 4D). After 12 days, the H3P$^+$ population was 3.5 times higher in FIJs-treated mice than control-treated mice (Fig 4E). In order to further confirm that proliferated population did indeed originate from CMs, double transgenic mice myh6-mER-Cre-mER/ZEG were generated with tamoxifen-induced $\alpha$-MHC promoter-driven EGFP to label adult CMs in green. After injection of adeno-Ctrl or adeno-FIJs directly into the adult hearts of M/Z mice and BrdU labeling for 4 days (Fig 4D), the adult CMs were isolated 12 days post-infection and the BrdU$^+$/EGFP$^+$ population was five times higher in FIJs-treated adult CMs than control-treated CMs (Fig 4F). The combination of these results shows that FIJs treatment could efficiently enhance adult CM proliferation *in vivo*.

## Delivery of gene cocktail FIJs enhances cardiomyocyte proliferation for improved heart function after myocardial infarction

The transcriptional expression pattern of FoxM1, Id1, and Jnk3 was also determined in the whole heart following myocardial infarction (MI) injury (Appendix Fig S5). The expression pattern of up-regulated FoxM1 and Id1 with down-regulated Jnk3 hints at their potential for heart regeneration after injury. The ability of FIJs treatment to enhance CM proliferation was demonstrated by direct injection into the heart after MI injury, as outlined in Fig 5A. Four days after adenoviral infection, the exogenous expression of FoxM1 or Id1 could be enhanced to more than two times that of the endogenous expression after MI, and Jnk3 had three times lower exogenous expression compared to the endogenous expression (Fig 5B). At the same time point, the BrdU$^+$ population, representing DNA synthesis, was five times higher in FIJs-treated mice than control-treated mice, and a twofold increase in the H3P$^+$ population was also detected (Fig 5C). Although it is possible that BrdU$^+$ cells may have

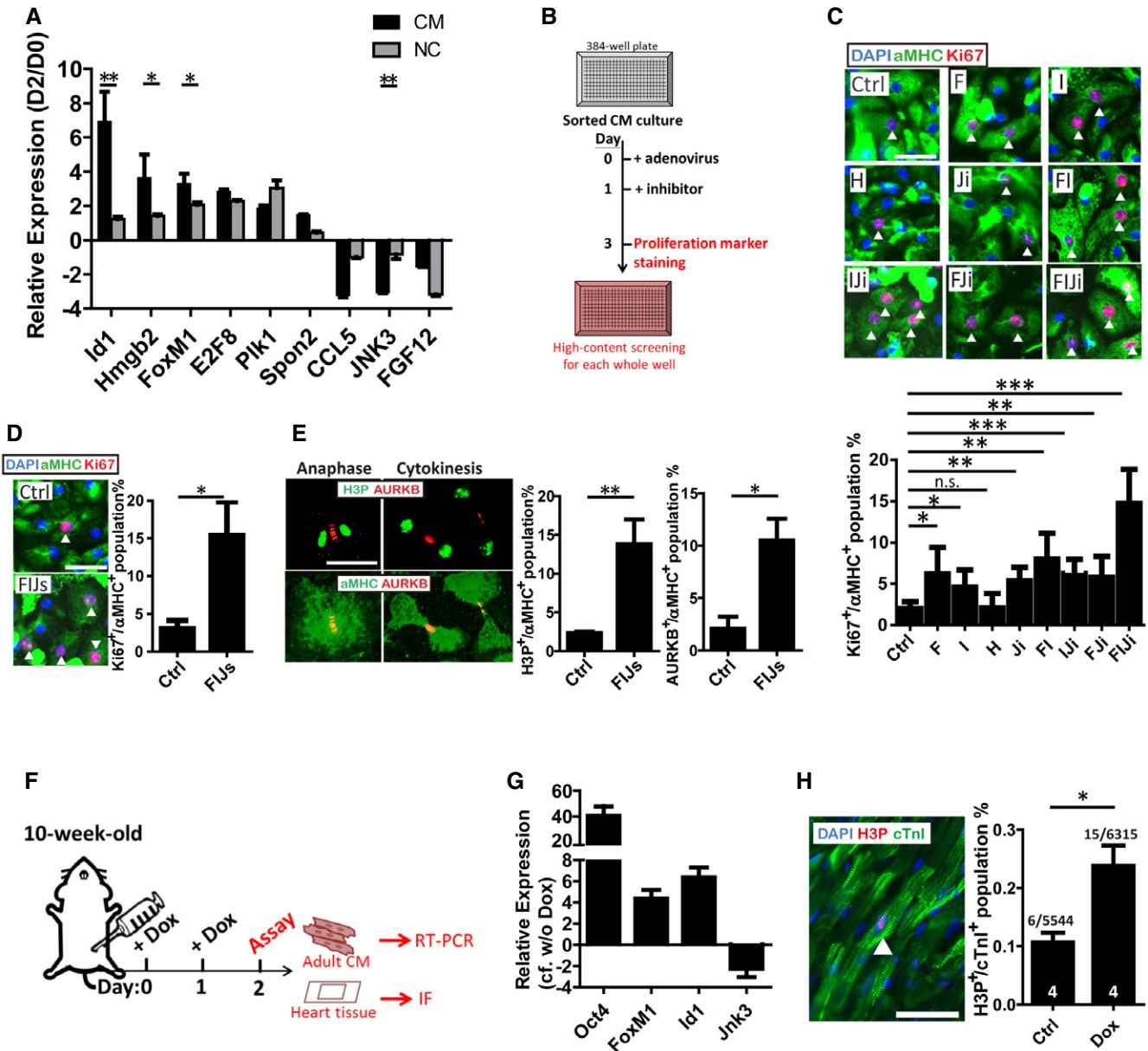

**Figure 3.  Gene screening for CM proliferation by mimicking early reprogramming.**

A  RNA expression ratio (D2-to-D0) of candidates selected from day 2 microarray data. *n* = 3.
B  Timeline followed for proliferation examination of neonatal CMs *in vitro*.
C  Ki67, α-MHC, and DAPI staining and quantification of CMs treated with control (Ctrl), or combinations of a FoxM1 up-regulation adenovirus (F), Id1 up-regulation adenovirus (I), Hmgb2 up-regulation adenovirus (H), or a small-molecule Jnk3 inhibitor (Ji). The arrowheads indicate Ki67⁺/α-MHC⁺ proliferated CMs. *n* = 3.
D  Ki67, α-MHC, and DAPI staining and quantification of CMs treated with Ctrl or combination of FI with Jnk3-shRNAs in place of the small molecule (Ji). The arrowheads indicate Ki67⁺/α-MHC⁺ proliferated CMs. *n* = 3.
E  Morphology and quantification of FIJs-treated CMs undergoing cytokinesis by H3P and Aurora kinase B (AURKB) staining. *n* = 3.
F  Timeline followed for adult reprogrammable mice administered the control or doxycycline *in vivo*.
G  RNA expression ratio of Oct4, FoxM1, Id1, or Jnk3 in doxycycline-injected versus control adult CMs isolated from reprogrammable mice. *n* = 3.
H  Immunofluorescence of H3P and cTnI and quantification of the heart tissue sections from control or doxycycline-injected mice. The arrowhead indicates H3P⁺/cTnI⁺ proliferated CMs. Sample size is indicated in the bar charts.

Data information: Scale bar represents 50 μm. Data are presented as mean ± SEM. Statistical analysis: unpaired *t*-test; *$P < 0.05$, **$P < 0.01$, and ***$P < 0.001$.

arisen from other cell populations, such as conversion of fibroblasts, the combination of these results shows that FIJs treatment could efficiently enhance CM proliferation after MI.

We considered whether this increase in CM proliferation would be met with a corresponding improvement in cardiac function, post-injury. Therefore, FIJs treatment or a control was injected into the

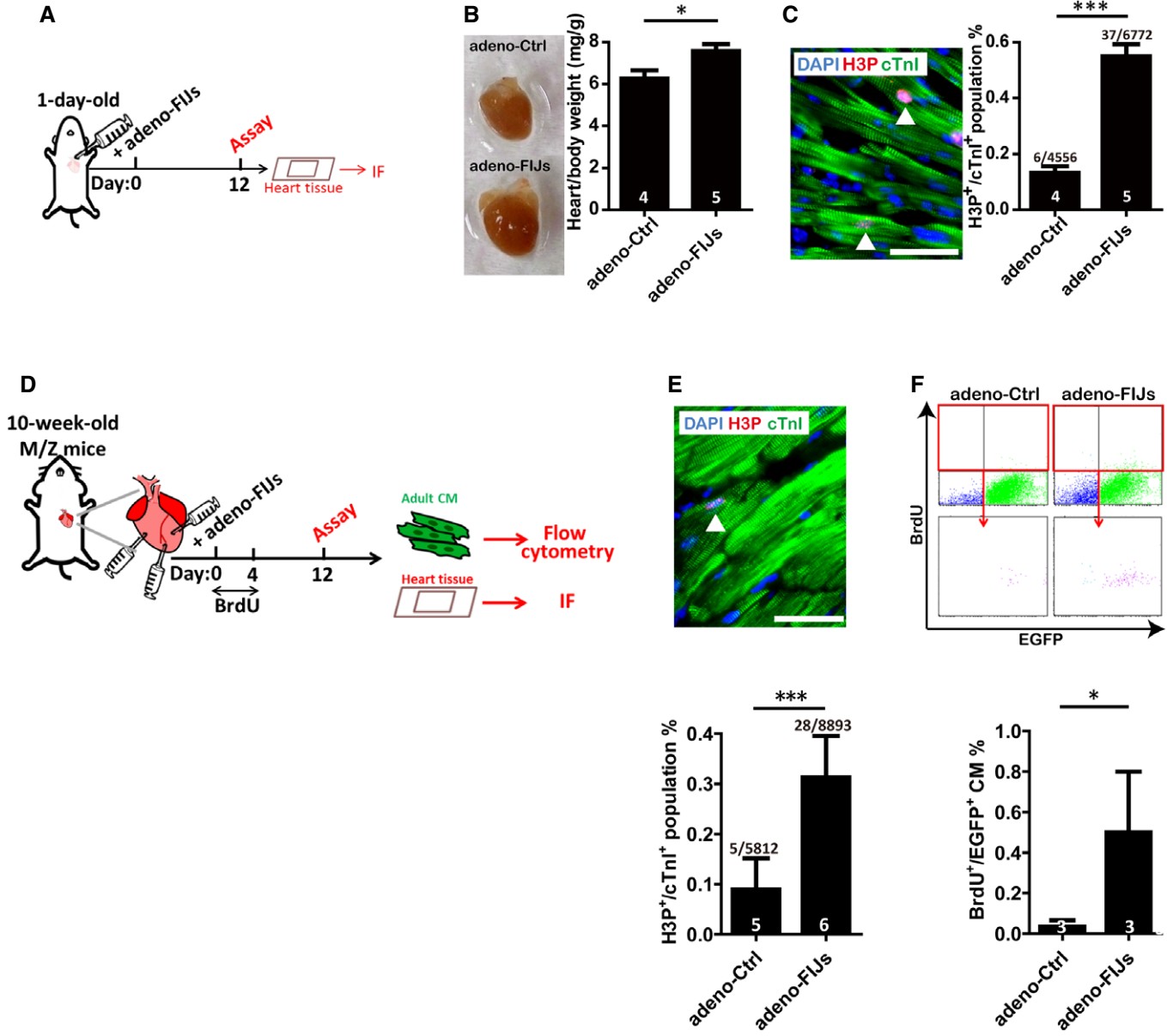

**Figure 4.   The defined gene cocktail induces CM proliferation in neonatal and adult mice.**

A   Timeline followed for neonatal P1 mice administered the Ctrl or FIJs.

B   The size of hearts isolated from Ctrl- or FIJs-treated mice; heart/body weight ratio of Ctrl- or FIJs-treated mice.

C   H3P, cTnI, and DAPI staining and quantification of heart tissues in Ctrl- or FIJs-treated mice. The arrowhead indicates H3P[+]/cTnI[+] proliferated CMs. Scale bar represents 50 μm.

D   Timeline followed for adult mice administered the adeno-Ctrl or adeno-FIJs.

E   Immunofluorescence of H3P and cTnI and quantification of the heart tissue sections from adeno-Ctrl or adeno-FIJs injected mice. The arrowhead indicates H3P[+]/cTnI[+] proliferated CMs. Scale bar represents 50 μm.

F   Quantification of BrdU[+]/EGFP[+] population % in isolated CMs from adeno-Ctrl or adeno-FIJs injected M/Z mice by flow cytometry.

Data information: Data are presented as mean ± SEM. Sample size is indicated in the bar charts. Statistical analysis: unpaired *t*-test; *$P < 0.05$, and ***$P < 0.001$.

heart immediately following MI injury and cardiac performance was measured by echocardiography (Fig 5A). The results showed a significantly higher ejection fraction and fraction shortening in FIJs-treated mice, representing overall improved cardiac function 21 days after MI injury (Fig 5D). At the same time, cardiac performance was also measured by catheterization, and the results showed a significantly higher dP/dt value in FIJs-treated mice under baseline condition. Higher ESPVR and PRSW and lower Tau value were confirmed in FIJs-treated mice under inferior vena cava occlusions, representing overall improved cardiac function 21 days after MI injury (Fig 5E). Furthermore, trichrome staining showed that FIJs-treated mice had a significantly smaller fibrotic area than

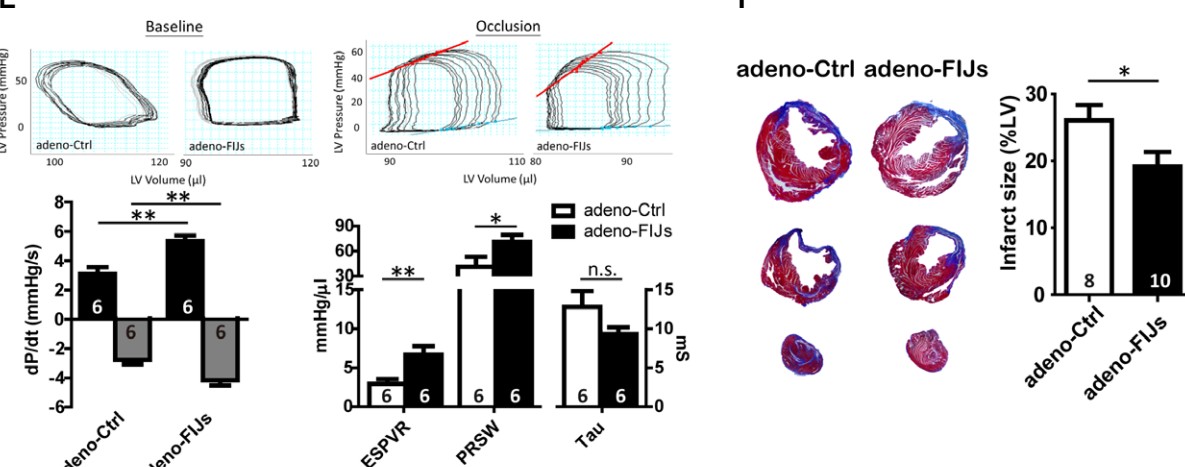

**Figure 5.    Enhancement of CM proliferation by delivery of a gene cocktail after MI.**

A    Timeline followed to analyze proliferation of CMs and heart improvement in mice subjected to MI followed by administration of FIJs treatment.

B    RNA expressional changes of FoxM1, Id1, and Jnk3 after adeno-Ctrl or adeno-FIJs treatment after MI. The dashed line represents endogenous gene expression level.

C    Immunofluorescence staining of BrdU or H3P and cTnI or EGFP and quantification in the heart tissues after MI. The arrowhead indicates cTnI$^+$/BrdU$^+$, cTnI$^+$/H3P$^+$ or EGFP$^+$/BrdU$^+$, EGFP$^+$/H3P$^+$ proliferated CMs. Scale bar represents 50 μm.

D    Ejection fraction and fraction shortening percentage in Ctrl- or FIJs-treated mice 21 days post-MI.

E    Catheterization in Ctrl- or FIJs-treated mice 21 days post-MI under baseline or inferior vena cava occlusion condition.

F    Trichrome staining and quantification showing percentage of fibrotic tissue (blue) in Ctrl- or FIJs-treated mice 21 days after MI.

Data information: Data are presented as mean ± SEM. Sample size is indicated in the bar charts. Statistical analysis: unpaired $t$-test; *$P < 0.05$, **$P < 0.01$ and n.s., no significance.

control-treated mice (Fig 5F). However, FIJs treatment causes only a modest increase in CM proliferation but results dramatic heart improvement (higher EF%, reduced fibrotic scarring, and higher dP/dt) after injury. We believe that there are likely some indirect or secondary effects such as changes to cardiomyocyte calcium cycling, adrenergic responsiveness, or afterload which contribute toward the overall effect on heart regeneration. Taken together, these results clearly indicate that FIJs treatment is able to enhance CM proliferation and improve cardiac performance after MI.

**Mechanisms of FoxM1, Id1, and Jnk3-shRNA that support cardiomyocyte proliferation**

In order to provide some insights into the mechanistic basis of FIJs activity, three mitosis markers, Aurkb, Mad2L1, and Plk1, were analyzed by RT–PCR following individual overexpression of FoxM1 and Id1, and knockdown of Jnk3. These markers all showed significantly increased expression after treatment with the individual components of FIJs, and even higher expression of these mitotic markers was observed after combined treatment (Fig 6A). This supports our previous data showing an increased proliferative CM population by H3P, Ki67, AURKB, and BrdU after FIJs treatment.

Since previous studies have shown that many of the factors supporting CM proliferation do so by regulating CDK inhibitors, to provide some mechanistic basis for the action of the FIJs treatment, we next looked at cyclin-dependent kinases and their inhibitors (Soonpaa *et al*, 1997; Pasumarthi *et al*, 2005; Mahmoud *et al*, 2013; Shapiro *et al*, 2014). We examined the impact of FIJs treatment on the expression of CDKs of the CM cell cycle (Fig 6B). The results showed that Cdk4 was up-regulated in Id1-treated CMs, whereas FoxM1 up-regulation and Jnk3-shRNA were most effective at increasing Cdk2 expression. Indeed, combined FIJs treatment effectively increased both Cdk4 and Cdk2 expression. In addition, each of the three components of the FIJs treatment was able to significantly enhance Cdk1 expression, and the combined FIJs treatment resulted in an even higher level of Cdk1 expression. Furthermore, to examine whether FoxM1, Id1, and Jnk3 expression levels played a role in the regulation of CDK inhibitors, the levels of p16, p21, and p27 were examined after FIJs treatments (Fig 6C). p16, as an inhibitor of CDK4, was down-regulated only in treatment groups containing Id1, and the expression of the CDK2 inhibitor p27 was more clearly reduced in CMs treated with Jnk3-shRNA. In addition, p21 was significantly inhibited by FoxM1. Although we note that the combined FIJs treatment with addition of Jnk3-shRNA might slightly compromise the effect of FoxM1 in terms of reducing p21 expression, the role of Jnk3-shRNA in decreasing p27 expression was important for higher Cdk2 expression after FIJs treatment. Figure 6D shows a schematic explaining the probable roles of each component of the FIJs treatment, although we do not rule out the possibility that other mechanisms are involved. We conclude that FIJs treatment is able to control CDK inhibitors for enhanced CDK expressions in order to drive increased CM proliferation—with Id1 inducing Cdk4 expression, FoxM1 and Jnk3-shRNA enhancing the Cdk2 expression, and the combination of all three significantly supporting cytokinesis. This three-factor combination, identified during early CM reprogramming, was able to increase CM proliferation *in vitro*, *in vivo*, and after cardiac injury, restoring cardiac performance after 21 days.

## Discussion

In this study, we demonstrated that there is strong up-regulation of mitosis-related genes during the second day of CM reprogramming. A cocktail of three genes, FoxM1, Id1, and Jnk3-shRNA (FIJs), derived from day 2 of CM reprogramming, induced CMs to re-enter the cell cycle and complete mitosis and cytokinesis. This gene cocktail was able to improve cardiac function after MI, thus showing great potential for heart regeneration.

Each gene in the FIJs cocktail was responsible for down-regulating different CDK inhibitors, therefore allowing CMs to re-enter the cell cycle (Fig 6). There is previous evidence to support the role of each gene in the cell cycle. FoxM1 has been previously reported to increase CM proliferation by down-regulating p21 or p27, but showed limited ability to promote heart recovery after injury (Bolte *et al*, 2012; Sengupta *et al*, 2013); Id1 has been claimed to prevent senescence via knockdown of p16 during *in vitro* culture of human fetal CMs (Ball & Levine, 2005). In our study, similar mechanistic results confirmed that FoxM1 down-regulated p27 to increase Cdk2 expression, and Id1 reduced p16 expression to support Cdk4 expression for the cell cycle (Fig 6). Jnk3 is predominantly expressed in brain, heart, and testis, but its role in the heart is not clearly described (Mielke, 2008). Jnk3 has been shown to stabilize p27 and led to growth arrest in C6 glioma cells, and so, following this precept, we confirmed that Jnk3-shRNA down-regulates p27, enhancing expression of Cdk2 in CMs for proliferation. Thus, each member of the gene cocktail is responsible for reducing specific CDK inhibitors and therefore supporting the CM cell cycle, as mentioned in previous studies. Most importantly, each gene independently up-regulated expression of Cdk1 and checkpoint genes for mitosis and cytokinesis in CMs (Fig 6). In our study, when treated with this three-gene cocktail, CMs not only re-entered the cell cycle but were also able to complete mitosis and cytokinesis (Fig 3). Compared to the previously mentioned single or double treatments, our gene cocktail changed expression of three specific genes to induce higher CM proliferation (seven times) which led to efficient improvement of heart function after injury. Although the effect on proliferation is modest overall (most < 1%), the combination of FIJs did show the potential for improvement of heart function after injury, as measured by several metrics including reduced fibrosis, increased ejection fraction measured by echocardiography, and increased dP/dt measured by catheterization. The next step would be to determine the concentration and ratio of FIJs cocktail to reach the optimized utilization to increase CM proliferation for heart regeneration after injury. We do not rule out the possibility that other mechanisms regulate this treatment in CMs, and the oncogenic activity of FIJs cocktail may be the limitation to compromise its clinical application. Therefore, the time point, period, and long-term effects of FIJs treatment should be determined to carefully evaluate its clinical application in the future.

In addition, microarray analysis at CM-D2 revealed up-regulation of some other cell cycle-related genes which have been previously shown to support CM proliferation, such as E2F1 and cyclin D1 (von Harsdorf *et al*, 1999; Tamamori-Adachi *et al*, 2003). This confirms that the second day of reprogramming is a useful time point at which to elucidate the underlying mechanisms of proliferative behavior in CMs. On the other hand, there were many other genes highlighted by our microarray at CM-D2 which were not directly

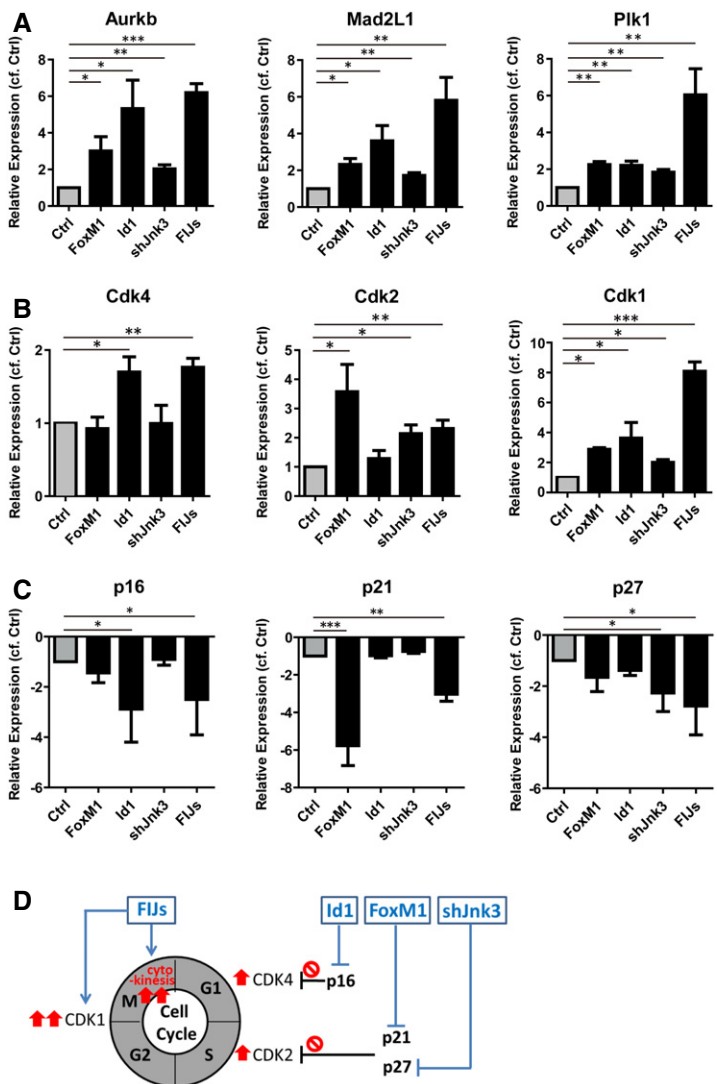

**Figure 6. Cell cycle effect of each gene or combination of the cocktail on CMs.**

A   RNA expression of mitosis checkpoint genes in CMs with FoxM1 or Id1 overexpression, Jnk3-shRNA, or combined treatment.
B   Transcriptional expression of cyclin-dependent kinases in FoxM1 or Id1 up-regulated, Jnk3 down-regulated, or combined treated CMs.
C   Real-time RT–PCR analysis of CDK inhibitor expression in CMs with FoxM1 or Id1 overexpression, Jnk3-shRNA, or combined treatment.
D   Schematic diagram demonstrating the proposed mechanism for FIJs-driven CM proliferation.

Data information: Data are presented as mean ± SEM. $n = 3$. Statistical analysis: unpaired $t$-test; $*P < 0.05$, $**P < 0.01$ and $***P < 0.001$.

associated with proliferation or the cell cycle, such as TXNIP. However, its role in heart improvement through down-regulation after ischemia–reperfusion insult has been confirmed to suppress mitochondrial function for anaerobic metabolism (Yoshioka *et al*, 2012). Also, expression of chemokines such as CCL5 showed significant repression in CMs, also aiding in the initiation of reprogramming at CM-D2 (Fig 2). The chemokine CCL5 has been previously shown to have a role in cardiac development, and down-regulation of CCL5 was found necessary for subsequent proliferation of epicardial cells (Velecela *et al*, 2013). Although the gene lists at CM reprogramming day 2 do not confirm their role in natural heart biology, they do show a high possibility of enhancing CM proliferation for efficient heart regeneration. In addition, the constituent genes in the FIJs cocktail were all confirmed to have expressional changes that

enhanced CM proliferation after MI injury (Appendix Fig S5). Thus, the exogenous expressional changes that we induced were to increase the effects of FIJs on enhanced CM proliferation for better heart functional improvement after injury, and our data as shown in Fig 5 support this. Also, the FIJs-treated proliferative CMs still showed clear sarcomeres, showing that these CMs did not change their significant characteristics. The similar progenitor state was described earlier that they established chemically induced cardiomyocyte-like cells from human somatic fibroblasts, and these cells were confirmed successfully engrafted into the hearts after transplantation (Cao *et al*, 2016). However, the progenitor state is unstable and is not easy to be controlled. In our study, we find the mitochondria or cardiac-related gene expression changes are not decreased at CM reprogramming day 2, but rather only start

decreasing at day 4 based on the Gene Ontology analysis (Fig 2C). FIJs was discovered from CM reprogramming day 2 when CMs still maintain their original characteristics but start to go back into cell cycle (Fig 2B and C). Therefore, we genetically hypothesized the method of cell fate alteration for novel approaches supporting cardiac regeneration, and the gene list derived from the second day of reprogramming may clue the way how CMs regain their proliferative capacity.

The second day of CM reprogramming, prior to the mesenchymal–epithelial transition (MET), is important time period. CMs must first change their original specific characteristics in order to regain proliferative capabilities. During the same stage of reprogramming, NCs showed increased sterol biosynthesis (Fig 2). In a previous study, murine embryonic fibroblasts could be transformed into sphere cells by suspension culture in lipid-rich medium (Rajanahalli *et al*, 2012). Therefore, we theorized that the period prior to the MET process may represent a critical period for the removal of cell identity and that changing patterns of gene expression may provide insights into trans-differentiation processes of different somatic cells.

This is the first study to screen the genes essential for CM proliferation based on the information derived from reprogramming, and the selected cocktail composed of FoxM1, Id1, and Jnk3-shRNA significantly improves heart growth and function after injury. Gene screening at the second day of reprogramming may reveal the key factors essential for manipulating cellular behavior which may be applied for improving poor tissue regeneration. Reprogramming assay not only shows the possibility of rejuvenation, but may also be further utilized for tissue regeneration through increasing understanding of the details of the process. In summary, this study has discovered a gene cocktail with significant cardiac regenerative effects, and the gene lists derived from CM reprogramming day 2 may provide useful insights into how CMs regain their proliferative capacity for heart regeneration. Furthermore, we hope that these findings will provide useful insight for further research into regeneration of other tissues.

# Materials and Methods

All animal experiments were conducted in accordance with the Guides for the Use and Care of Laboratory Animals (ARRIVE guidelines), and all of the animal protocols have been approved by the Experimental Animal Committee, Academia Sinica, Taiwan. Normal C57BL/6 (male) mice were housed in individually ventilated cages (IVCs) system in animal core facility at Academia Sinica.

### Isolation of cardiomyocytes and non-cardiomyocytes from neonatal heart

Cardiomyocytes (CMs) and non-cardiomyocytes (NCs) were isolated from 2- or 3-day-old C57BL/6 mice using a previously described protocol, with some modifications (Condorelli *et al*, 2002). In brief, heart tissue was minced and digested by 1 mg/ml trypsin (Sigma-Aldrich) at 4°C for 2 h after removal of the atrium and aorta. The minced tissues were treated with 0.8 mg/ml collagenase II (Invitrogen) at 37°C for 15 min, and the cells were collected after filtering with a 40-μm strainer for clearance of debris.

### Reprogramming procedure

The reprogramming experiment was modified from a standard procedure (Takahashi & Yamanaka, 2006) as follows: The concentration of doxycycline was 1 μg/ml for inducing reprogramming of cells isolated from OSKM mice. The iPSC medium was GMEM basal medium with 15% FBS.

### Transcriptomic analysis

Samples from different reprogramming time points were hybridized to a Mouse Oligo Microarray (Agilent) following the manufacturer's procedure, and arrays were scanned with Microarray Scanner System (Agilent). All 23 CEL files were analyzed by GeneSpring GX software (Agilent) with quantile normalization and median polish probe summarization using D0 as a baseline. The expression levels in the first quantile were filtered out to remove noise, and the gene that was not detected in at least two of the three biological replicates was further removed for additional analysis. Genes were defined as differentially expressed if they had fold changes of at least $\pm 2$ combined with the Student's *t*-test ($P < 0.05$) with the Benjamini–Hochberg adjustment for false discovery rate (FDR). Gene Ontology analysis was conducted using DAVID software (Huang *et al*, 2009). Besides two replicates for reprogramming day 6 of NC, the biological replicates were three for different time points during cardiomyocyte (CM) or non-cardiomyocyte (NC) reprogramming. The accession number for microarray data is E-MTAB-5295.

### Treatment of JNK3 inhibitor XII, SR-3576

JNK3 inhibitor XII, SR-3576 (Millipore), specific to JNK3 was administrated as 1 μmol/l to neonatal CMs *in vitro* for 3 h.

### Production and purification of recombinant adenoviral vectors

Foxm1, Id1, or Hmgb2 cDNA were amplified from total reverse-transcriptized cDNA purified from C57BL/6 mice with Phusion High-Fidelity PCR Master Mix (New England Biolabs). The primers for amplification are listed in Appendix Table S2. The amplified fragment was cloned into the site next to ires-EGFP of pENTR plasmid. These specific gene-carrying pENTR plasmids were then recombined into pAd/PL-DEST plasmids using a pAd/PL-DEST Gateway Vector Kit (Invitrogen). Viral condensation was completed by $CsCl_2$ gradient centrifugation.

### Injection of adenoviral vectors in neonatal and adult mice

For neonatal mice, adenoviral infection of hearts has been previously reported in detail (Christensen *et al*, 2000; Ebelt *et al*, 2008). One-day-old mice were anaesthetized by cooling on ice for 2 min and were injected into the thoracic cavity at the left parasternal position with a Hamilton syringe with 30-gauge needle then returned to their mothers feeding for 12 days. The mice were then sacrificed, and hearts were collected for subsequent experiments.

For adult mice, intracardiac injection was performed at the dose of $1 \times 10^{11}$ viral particles per mouse. For therapy, adenovirus was injected to the border zone of the injured heart at three sites immediately following myocardial infarction.

## Immunofluorescence

The cells were fixed with 4% paraformaldehyde and permeabilized by 0.3% Triton X-100 in blocking buffer with 5% goat serum in PBS for 1 h. Cells were then stained with primary antibodies, anti-α-sarcomeric actinin (Sigma), anti-cardiac troponin T (DSHB), anti-cardiac troponin I (DSHB), and anti-α-MHC (Abcam) for CM staining; anti-vimentin (Santa-Cruz) for CF staining. Ki-67 (Genetex), histone H3 phosphorylated at serine 10 (Millipore), Aurora B kinase (Abcam) for gene screening. Secondary antibodies conjugated with Alexa fluor-488 or Alexa fluor-568 (Life Technology) were incubated for 1 h at room temperature after washing with PBS three times, and the nuclei were stained with DAPI (Life Technologies) for 5 min.

The tissue sections were deparaffinized, rehydrated, and antigens retrieved by boiling twice in sodium citrate solution. Then, the sections underwent the same immunofluorescence procedure as mentioned previously.

## Quantitative real-time PCR

Reverse transcription was performed following the protocol of SuperScript III Reverse-Transcriptase kit (Life Technology). SYBR Green Real-Time PCR master mix was used to quantify expression of each gene, with GAPDH used for normalization. The primers used for quantification are listed in Appendix Table S3.

## Efficiency of recombinant adenoviral infection *in vitro* and *in vivo*

These specific gene-carrying pAd/PL-DEST plasmids were designed to produce adenovirus with specific gene and EGFP expression following a previously described procedure (Luo *et al*, 2007). The infection efficiency was more than 90%, and the adenovirus-infected CMs with EGFP fluorescence were confirmed to express both FoxM1 and ID1 (Appendix Fig S6A). Overexpression of each specific gene was confirmed to be > 100-fold. For *in vivo* infection, the exogenous expression of FoxM1 or Id1 could be enhanced more than 2 times that of the endogenous expression after MI, and Jnk3 had three times lower exogenous expression than endogenous expression 4 days after adenoviral infection (Fig 5B). At the same time, EGFP expression in the heart tissues was confirmed localized to the area surrounding the adenovirus injection site near the injury site and co-localized with FoxM1 or ID1 expression (Appendix Fig S6B). However, after 7 days the induction vanished and FIJ expression was at the same level in the adeno-Ctrl- or adeno-FIJs-treated heart tissues (Fig 5B).

## Adult cardiomyocyte isolation

Adult ventricular CMs were isolated from male C57BL/6 mice on a Langendorff apparatus. The hearts were removed from the anaesthetized mice after heparinization for 10 mins, and then were cannulated for retrograde perfusion with $Ca^{2+}$-free Tyrode solution (NaCl 120.4 mmol/l, KCl 14.7 mmol/l, $KH_2PO_4$ 0.6 mmol/l, $Na_2HPO_4$ 0.6 mmol/l, $MgSO_4$ 1.2 mmol/l, HEPES 1.2 mmol/l, $NaHCO_3$ 4.6 mmol/l, taurine 30 mmol/l, BDM 10 mmol/l, glucose 5.5 mmol/l). After 3-min of perfusion, the solution mixed with

$Ca^{2+}$-free Tyrode solution supplemented with collagenase B (0.4 mg/g body weight, Roche), collagenase D (0.3 mg/g body weight, Roche), and protease type XIV (0.05 mg/g body weight, Sigma-Aldrich) was used to digest the hearts. After digestion, the ventricles were cut from the cannula and teased into small pieces in the digestion solution neutralized by the $Ca^{2+}$-free Tyrode solution containing 10% FBS. Adult CMs were dissociated from the digested tissues by gentle pipetting and collected after removing the undigested tissues by filtering through a nylon mesh of 100 μm pore size.

## Myocardial infarction

C57BL/6 mice (10 weeks old, male) were randomized and anesthetized by isofluorane inhalation, endotracheally intubated, and placed onto a rodent ventilator. The left anterior descending (LAD) coronary artery was visualized and occluded with a prolene suture after first removing the pericardium. The whitening of a region of the left ventricle was confirmed immediately post-ligation as successful myocardial infarction.

## Echocardiography

Transthoracic two-dimensional echocardiography was analyzed using a Vivid-q Ultrasound (General Electric Company) equipped with a 5.0–13.0 MHz intraoperative probe. M-mode tracings in parasternal short-axis view were blinded measure left ventricular anterior and posterior wall thickness, and the internal diameters at end-systole and end-diastole were used to calculate the left ventricular fractional shortening and ejection fraction.

## Catheterization

The hemodynamics was measured by catheterization with a pressure sensor diameter of 1.4F and 0.8F along the catheter body (Millar Instruments, Houston, SPR-839) after 21 days after surgery. The catheter was inserted into the right carotid artery and advanced to the left ventricle (LV), and one milliliter of blood was drawn from the heart to calibrate the electrical conductivity for volume conversion. Baseline LV pressure-volume loops were recorded after stabilization, and the inferior vena cava was transiently compressed through an incision in the upper abdomen to change preload. The catheterization data with volume calibration were analyzed with commercial software (PVAN3.2; Millar Instruments).

## Measurement of infarct size

The infarct and the total area of the sections were traced in the digital images and measured by AxioVision software. Total area of a section was measured, and the infarct size was calculated and expressed as a percentage. Three serial sections were analyzed for each heart, and eight to ten hearts for each group were measured for statistics.

## Cardiomyocyte quantification

For gene screening, cells were stained with Ki67 or H3P and DAPI and image acquisition of each whole well was performed by

**The paper explained**

**Problem**

Although remnant cardiomyocytes (CMs) possess a certain degree of proliferative ability, efficiency is too low for cardiac regeneration after injury.

**Results**

Based on analysis within the initiation phase of CM reprogramming, we identified a cocktail of three genes, FoxM1, Id1, and Jnk3-shRNA (FIJs), induced CMs to re-enter the cell cycle with complete mitosis and cytokinesis, and significantly improved cardiac function and reduced fibrosis after myocardial infarction.

**Impact**

Our findings present a cocktail FIJs that may be useful in cardiac regeneration, and also provide a practical strategy for probing reprogramming assays for regeneration of other tissues.

ImageXpress Micro High-content screening microscope (Molecular Devices). The derived images were further utilized for evaluating the percentage of proliferative CMs. Ki67$^+$ or H3P$^+$ CMs were further evaluated by MetaMorph Image Analysis Software with manual verification.

To quantify cells from tissue sections, cell number was determined by quantifying five sections from each heart in the border zone around the infarct area and three images were randomized, blinded, and taken at a magnification of 200× for each section. More than 1,000 cell numbers of CMs were counted for evaluating percentage of proliferative CMs from the tissue sections of each heart. The cells from tissue sections were stained with BrdU or H3P and cTnI to evaluate proliferative CMs *in vivo*.

**Statistics**

All statistical data were analyzed in GraphPad Prism and shown as mean ± standard error of the mean (SEM). Unpaired Student's *t*-test and two-way ANOVA were applied for statistical comparisons, and a *P*-value of < 0.05 was considered significant. The exact *P*-values in each figure were listed in Appendix Table S4.

**Expanded View** for this article is available online.

**Acknowledgements**

The authors are grateful to Dr. Shu-Jen Chou (Institute of Plant and Microbial Biology, Academia Sinica) for assistance with microarray analysis; Dr. Darrell N. Kotton (Center for Regenerative Medicine (CReM), Boston University) for the gift of a single lentiviral stem cell cassette and Dr. Kenneth D. Poss (Howard Hughes Medical Institute, Duke University) and Dr. Huck-Hui Ng (Genome Institute of Singapore) for the suggestions and advice for this project. FACS, high-content imaging, and animal studies were supported by Core Facility located at the Institute of Biomedical Sciences, Academia Sinica. RNAi reagents were obtained from the National RNAi Core Facility located at the Institute of Molecular Biology/Genomics Research Center, Academia Sinica, supported by the National Core Facility Program for Biotechnology Grants of NSC (NSC 100-2319-B-001-002). This study was supported by the Ministry of Science and Technology (MOST 102-2321-B-001-069-MY3 and 104-2325-B-001-010) and the Academia Sinica Translational Medicine Program.

**Author contributions**

PCHH conceived the project, developed the hypothesis and designed the study; Y-YC performed experiments and analyzed data; Y-TY analyzed data and provided input into interpretation; Y-YC and DJL wrote the paper; AHAL, Y-PW, S-CR, and P-JL performed the animal surgery, echocardiography, and catheterization.

**Conflict of interest**

The authors declare that they have no conflict of interest.

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
