## [Review Process File · EMBO Molecular Medicine]

Reprogramming-derived Gene Cocktail Increases Cardiomyocyte Proliferation for Heart Regeneration

Yuan-Yuan Cheng, Yu-Ting Yan, David J. Lundy, Annie H.A. Lo, Yu-Ping Wang, Shu-Chian Ruan, Po-Ju Lin and Patrick C.H. Hsieh

Corresponding author: Patrick Hsieh, Academia Sinica

Review timeline:

Submission date:	28 April 2016
Editorial Decision:	16 June 2016
Revision received:	27 September 2016
Editorial Decision:	22 November 2016
Revision received:	27 November 2016
Accepted:	29 November 2016

Transaction Report:

Editor: Céline Carret

1st Editorial Decision

16 June 2016

Thank you for the submission of your manuscript to EMBO Molecular Medicine. We have now heard back from the three referees whom we asked to evaluate your manuscript. Although the referees find the study to be of potential interest, they also raise a number of concerns that must be thoroughly addressed for the paper to be considered further.

You will see that while all three referees found the study novel and interesting, they also all agree that more data are needed to make it more robust and conclusive (see ref2 and 3 especially).

I would like to give you the opportunity to revise your manuscript, with the understanding that the referees concerns must be all fully addressed and experimentally when needed and that acceptance of the manuscript would entail a second round of review. Please note that it is EMBO Molecular Medicine policy to allow only a single round of revision and that, as acceptance or rejection of the manuscript will depend on another round of review, your responses should be as complete as possible.

Revised manuscripts should be submitted within three months of a request for revision; they will otherwise be treated as new submissions, except under exceptional circumstances in which a short extension is obtained from the editor. Please make sure to follow our guidelines to revise your article and provide it in the right format.

I look forward to seeing a revised form of your manuscript as soon as possible.

***** Reviewer's comments *****

Referee #1 (Remarks):

This is an interesting manuscript from a respected group of cardiovascular scientists that identifies key molecular determinants for reactivating the proliferative capacity of adult cardiomyocytes both *in vitro* and *in vivo*. The experimental strategy is novel, based upon the ability of cardiomyocytes to be converted into iPS cells with the Yamanaka factors, and then identifying a transitional phase prior to their full conversion to pluripotency where the cells start dividing at a time when the cardiomyocyte phenotype is not completely lost. The ability to convert the cardiomyocytes to iPS directly *in vitro* has previously shown to be challenging and there is always a question as to whether the iPS derived cells originated from authentic cardiomyocytes or non-cardiomyocytes that are contaminating and actually would most likely preferentially be transformed to the iPS cell fate. To circumvent this problem, the authors generated a transgenic, inducible line where the Yamanaka factors can be exclusively activated in the cardiac muscle cells following their isolation and identification by GFP reporters, an elegant approach. Subsequently, this transitional state is characterized by transcriptional profiling and direct comparison to non cardiomyocytes and candidates were identified and tested *in vitro* in various combinations to determine their effects *in vitro* on adult cardiomyocyte proliferation and cytokinesis, leading to the identification of three factors that can reactivate cell proliferation *in vitro* and *in vivo*, which is shown using adenoviral vectors. Additional circumstantial evidence is provided by examining the time course of the three gene candidates in model systems where there is a transition towards the loss of proliferative capacity. As a final proof, the authors go on to show that these adenoviral vectors can increase both the incidence of cardiomyocyte proliferation, as well as global increases in EF. *In toto*, this is an impressive, novel study that should be of considerable interest to the CV community, and, in my view, is of sufficient interest to justify publication in EMM. There are a few points below that should be addressed.

- 1) The effect of the three genes on *in vivo* proliferation appears rather modest, in most cases less than 1 percent. How does this compare to the other main signaling pathway that has recently been uncovered by multiple groups, the neuregulin pathway?
- 2) Given the relatively modest effect on the proliferative capacity, it is a bit surprising that there is a much larger effect on global EF post MI. This raises the issue of a secondary effect of some sort, e.g., calcium cycling, adrenergic responsiveness, afterload, etc. This should be mentioned.
- 3) Another group used a similar approach to generate iPS from cardiomyocytes earlier () which should be referenced
- 4) Is there any evidence that the state created reflects a return to a "progenitor-like" state? A similar approach using small molecules has been shown by the Sheng Ding lab in Science....should be referenced and results herein compared to

Referee #2 (Remarks):

In this manuscript, Cheng et al. first profile expression changes during the reprogramming process from cardiomyocytes or cardiac fibroblasts to iPS cells. Then they used this gene profiling data to identify candidates that have the potential to promote cardiomyocyte proliferation. The authors found a cocktail, including FoxM1, Id1 and Jnk3-shRNA (FIJs), that promotes cardiomyocyte proliferation, both *in vivo* and *in vitro*, and improves heart function after myocardial infarction. The strength of this work is the novelty of the concept and approach that they used to find targets for heart regeneration. The microarray data they generated will be a valuable resource for this field. The major weakness of this work is that the FIJs cocktail they identified has limited improvement for cardiac function and limited therapeutic implications because of the oncogenic activity of the cocktail. Overall, I suggest a revision for this paper. The following are the itemized comments:

1. Page7, line19. The authors identified expression changes in over 15,000 genes. There are only 20,000 genes in mouse genome. Does that mean almost every gene changed its expression during this process? Will that compromise the importance of this gene profiling?
2. Figure 1. What's the purity of cardiomyocytes after culturing for 6 days? Since this work relies on high purity of CM and non-CM, the authors should check the cell purity at different time point by FACS.
3. According to the authors' theory (Figure 2F), the CM adopts higher proliferation rate but still maintains CM identity in day2. The authors should provide data to support this claim.

4. More quantitative methods (such as FACS analysis) should be used to quantify the percentage of proliferating cells.
5. Figure5c. The authors should demonstrate that the proliferating CM are the one overexpressing FIJs.
6. Figure5d. The authors should use cardiac MRI or cardiac catheterization to measure EF. Echocardiography is not a quantitative way for measuring EF. What about FS? The improvement of cardiac function after FIJs administration seems very mild. The authors should rephrase their statement in Page2 line 19.
7. FoxM1 and Id1 are well known oncogenes, and Jnk3 is a tumor suppressor. The oncogenic activity of this cocktail may compromise its clinical potential. The authors should discuss on this possibility.
8. - Supp Fig3, the down-regulation of Id1 and the up-regulation of Jnk3 expression do not coincide with the loss of CM proliferation.

Referee #3 (Comments on Novelty/Model System):

Inadequate because although model system used is acceptable, more rigorous model systems are available to be more confident of true cell division.

Referee #3 (Remarks):

Cheng et al. report that a cocktail of three genes discovered from reprogramming of cardiomyocytes toward iPSCs using OSKM was efficient in stimulating cardiomyocyte proliferation in vitro and in vivo. Based on the gene changes at day 2 when cardiomyocytes proliferate before starting to change fate, they focused on three specific genes that are then used throughout for in vitro and in vivo studies. While the work is interesting and based on a novel concept, there are several limitations which preclude confidence in the conclusions made based on the data.

1. The in vitro study for cardiomyocyte proliferation was conducted in day 1-3 neonatal cardiomyocytes which already have a relatively high baseline of cell proliferation as indicated in Figure 3D and E. The control has least 3-5% proliferating cells, which goes up to 10-15% with the FIJs cocktail. It seems preferable to use more mature cardiomyocytes to conduct such experiments, e.g. day 7-10 cardiomyocyte.
2. The authors used end point staining for Ki67, phospho histone H3 and Aroua Kinase B as proof of cardiomyocyte cell division in vitro. Since each marker has its limitations on truly interpreting the presence of cell division, it would be useful if the authors showed time lapse video of the cell division and how many cells undergo cytokinesis and whether the cells survive or not after division and for how long.
3. The in vivo study did not show convincing evidence that there is proliferation resulting in mature functional cardiomyocytes and there is a need to use lineage tracing mouse models to prove this finding. For example, mosaic analysis of dual marker (MADM) mice (Ali et al., PNAS. 2014;111(24):8850-5) or some such system would be much more rigorous, particularly given the notorious difficulty in ensuring that cell division of cardiomyocytes actually occurred in vivo.
4. The authors should show the efficiency of overexpressing Id1 and FoxM1 and knockdown of JNK3 in vitro and in vivo on protein level using western blot or immunofluorescence. It is mentioned in the methods section and qPCR data in vivo is shown in Figure 5B.
5. In figure 5D, it is surprising that the ejection fraction after 1 day went dramatically down and stayed at the same value after 3 weeks. Usually it takes 3-7 days for the scar to stabilize so the function stays partially preserved at day 1 and then there is typically progressive dysfunction after 3 weeks.
6. The in vivo study in Figure 5 shows very subtle improvement in cardiac function from ~35% to ~40% (Figure 5D) and based on that subtle difference the authors claimed that there is very subtle induction in proliferation (3 to 8 H3P+ cells out of ~4000) which will potentially treat heart failure.

The current data are not convincing and the authors should track the proliferative cells *in vivo* and determine what fate they take. Are these cells from cardiomyocytes that divided or are they from resident cardiac progenitors, if any, that divided?

7. The *in vivo* study was conducted for only 3 weeks and usually to obtain chronic heart failure 8-12 weeks are needed. 3 weeks is too short and is insufficient time to make conclusions.

1st Revision - authors' response

27 September 2016

Reviewer #1

This is an interesting manuscript from a respected group of cardiovascular scientists that identifies key molecular determinants for reactivating the proliferative capacity of adult cardiomyocytes both *in vitro* and *in vivo*. The experimental strategy is novel, based upon the ability of cardiomyocytes to be converted into iPS cells with the Yamanaka factors, and then identifying a transitional phase prior to their full conversion to pluripotency where the cells start dividing at a time when the cardiomyocyte phenotype is not completely lost. The ability to convert the cardiomyocytes to iPS directly *in vitro* has previously shown to be challenging and there is always a question as to whether the iPS derived cells originated from authentic cardiomyocytes or non-cardiomyocytes that are contaminating and actually would most likely preferentially be transformed to the iPS cell fate. To circumvent this problem, the authors generated a transgenic, inducible line where the Yamanaka factors can be exclusively activated in the cardiac muscle cells following their isolation and identification by GFP reporters, an elegant approach. Subsequently, this transitional state is characterized by transcriptional profiling and direct comparison to non cardiomyocytes and candidates were identified and tested *in vitro* in various combinations to determine their effects *in vitro* on adult cardiomyocyte proliferation and cytokinesis, leading to the identification of three factors that can reactivate cell proliferation *in vitro* and *in vivo*, which is shown using adenoviral vectors. Additional circumstantial evidence is provided by examining the time course of the three gene candidates in model systems where there is a transition towards the loss of proliferative capacity. As a final proof, the authors go on to show that these adenoviral vectors can increase both the incidence of cardiomyocyte proliferation, as well as global increases in EF. *In toto*, this is an impressive, novel study that should be of considerable interest to the CV community, and, in my view, is of sufficient interest to justify publication in EMM.

We thank the reviewer for the complementary remarks.

1) The effect of the three genes on *in vivo* proliferation appears rather modest, in most cases less than 1 percent. How does this compare to the other main signaling pathway that has recently been uncovered by multiple groups, the neuregulin pathway?

We thank the reviewer for this question. We agree that the effect on proliferation is modest overall. However, we have shown that this combination did improve heart function by several different metrics following injury including reduced fibrosis and increased ejection fraction measured by echocardiography. In this revision, we are pleased to provide data obtained by catheterization which showed functional cardiac improvements (Figure 5E) in line with our previous findings.

With regards to Neuregulin, although Neuregulin can induce cardiomyocyte numbers more than 1 %, the baseline of the proliferating adult cardiomyocytes *in vivo* in the Bersell et al. study was higher than the adult cardiomyocytes in our study (Polizzotti, Ganapathy et al. 2015). Also, Bersell et al. examined *in vivo* effects by systemic injection of neuregulin recombinant protein and measured BrdU levels over 9 days. In our study, FIJs treatment was locally injected direct into the heart and BrdU was assessed over a 4 day period. We find it difficult to compare the efficacy of these two studies, given the different delivery methods, different cell types, and different metrics taken. Out to interest, neuregulin 1 was also listed in our data, showing 3.2 times overexpression during the second day of CM reprogramming, as shown in appendix table 1. We have updated the Discussion section of the manuscript to reflect this information, as follows:

Discussion

Page: 13

Compared to the previously mentioned single or double treatments, our gene cocktail changed expression of 3 specific genes to induce higher CM proliferation (7 times) which led to efficient improvement of heart function after injury. Although the effect on proliferation is modest overall (most less than 1 %), the combination of FIJs did show the potential for improvement of heart function after injury, as measured by several metrics including reduced fibrosis, increased ejection fraction measured by echocardiography, and increased dP/dt measured by catheterization. The next step would be to determine the concentration and ratio of FIJs cocktail to reach the optimized utilization to increase CM proliferation for heart regeneration after injury. We do not rule out the possibility that other mechanisms regulate this treatment in CMs, and the oncogenic activity of FIJs cocktail may be the limitation to compromise its clinical application. Therefore, the time point, period, and long-term effects of FIJs treatment should be determined to carefully evaluate its clinical application in the future.

2) Given the relatively modest effect on the proliferative capacity, it is a bit surprising that there is a much larger effect on global EF post MI. This raises the issue of a secondary effect of some sort, e.g., calcium cycling, adrenergic responsiveness, afterload, etc. This should be mentioned.

We agree that the small increase in proliferation is unlikely to be solely responsible for the morphological and functional improvements we have observed. We have also confirmed these functional improvements by catheterization, as shown in Figure 5E. In terms of secondary effects, we have found that FIJs-treated cardiac fibroblasts showed decreased fibrotic gene expression, and masson's trichrome staining revealed reduced fibrosis in FIJs-treated hearts following MI. We have included a sentence in the Results section of the manuscript to address this point.

Results

Page: 10

Delivery of Gene Cocktail FIJs Enhances Cardiomyocyte Proliferation for Improved Heart Function after Myocardial Infarction

However, FIJs treatment causes only a modest increase in CM proliferation but results dramatic heart improvement (higher EF%, reduced fibrotic scarring, higher dP/dt) after injury. We believe that there are likely some indirect or secondary effects such as changes to cardiomyocyte calcium cycling, adrenergic responsiveness, or afterload which contribute towards the overall effect on heart regeneration.

3) Another group used a similar approach to generate iPS from cardiomyocytes earlier () which should be referenced

We thank the reviewer for reminding us of this publication. We have included two additional references related to reprogramming CMs into iPSCs in the Introduction section of the manuscript, as follows:

Introduction

Page: 4

Although the reprogramming of CMs into iPSCs has been reported (Rizzi, Di Pasquale et al. 2012, Xu, Yi et al. 2012), it is still unclear whether CMs, which have poor proliferative capacity, undergo similar reprogramming processes to MEFs, which have a high proliferation rate.

4) Is there any evidence that the state created reflects a return to a "progenitor-like" state? A similar approach using small molecules has been shown by the Sheng Ding lab in Science....should be referenced and results herein compared to.

We thank the reviewer for these comments. The chemically induced cardiomyocyte-like cells (ciCMs) discovered by the Sheng Ding's lab showed functional sarcomeres and contraction, and those cells could be successfully engrafted into the heart after transplantation (Cao, Huang et al. 2016). However, one concern may be that the epigenetic memory of the somatic fibroblasts may lead the ciCMs to be unstable at the progenitor state. In our study, we find the mitochondria or cardiac-related gene expression changes are not decreased at CM reprogramming day 2, but rather only start decreasing at day 4 based on the Gene Ontology analysis (Fig 2C). Therefore, FIJs was discovered from CM reprogramming day 2 when CMs still maintain their original characteristics but start to go back into cell cycle (Fig 2B-C). We appreciate that the reviewer has given these

comments, and we have included this information in the Discussion section of the manuscript, as follows:

Discussion

Page: 14-15

The similar progenitor state was described earlier that they established chemically induced cardiomyocyte-like cells from human somatic fibroblasts, and these cells were confirmed successfully engrafted into the hearts after transplantation. However, the progenitor state is unstable and is not easy to be controlled. In our study, we find the mitochondria or cardiac-related gene expression changes are not decreased at CM reprogramming day 2, but rather only start decreasing at day 4 based on the Gene Ontology analysis (Fig 2C). FIJs was discovered from CM reprogramming day 2 when CMs still maintain their original characteristics but start to go back into cell cycle (Fig 2B-C).

Reviewer #2

1. Page7, line19. The authors identified expression changes in over 15,000 genes. There are only 20,000 genes in mouse genome. Does that mean almost every gene changed its expression during this process? Will that compromise the importance of this gene profiling?

We apologize for the misunderstanding. We meant that we measured the expression levels of over 15,000 genes, not that all of them were changed during the early reprogramming process. We have re-written the sentence in question to provide better clarity:

Results

Page: 5

Cardiomyocytes regain proliferative capabilities during early reprogramming
We examined expression levels of over 15000 genes during the time period measured.

2. Figure 1. What's the purity of cardiomyocytes after culturing for 6 days? Since this work relies on high purity of CM and non-CM, the authors should check the cell purity at different time point by FACS.

We agree with the reviewer that the purity of CMs is critically important for this study. CMs were purified by mitochondrial dye TMRM staining with fluorescence-activated cell sorting (FACS), which have previously shown to obtain > 95% purity (Hattori, Chen et al. 2010). In our study, we further use a double-gated criterion to purify cardiomyocytes based on TMRM as well as cell size and mass (Fig 1A). However, since the number of the isolated neonatal CMs is not enough for repeated FACS analysis at later time points, we instead measured the purity of CMs by measuring with high-content microscopy to confirm that each cell expresses cTnI after culturing for 2 days or 6 days, as we described in Material and Methods section. The results show that every TMRM⁺⁺ cell (thus identified as a CM) expresses cTnI with clear sarcomeres after culturing *in vitro* for 6 days as shown in Fig 1B. On the contrary, TMRM⁺ cells (identified as NCs) show positive vimentin staining.

Figure and Figure legends

Page: 28

Figure 1. Early reprogramming process of CMs and NCs.

B Cardiac troponin I (cTnI) and α -Sarcomeric actinin (α -SA) staining of the TMRM⁺⁺ population (CMs) and Vimentin staining of the TMRM⁺ population (NCs).

3. According to the authors' theory (Figure 2F), the CM adopts higher proliferation rate but still maintains CM identity in day2. The authors should provide data to support this claim.

We thank the reviewer for the comments. Based on the microarray data shown in Fig 2B, the expression profiles were divided into 8 clusters based on the different expression patterns observed during early reprogramming, and mitochondria or cardiac-related genes were focused on cluster VII. Most cardiac-related gene expression levels are not decreased at CM reprogramming day 2, but rather only start decreasing at day 4. Also, based on the Gene Ontology analysis (Fig 2C), we find the mitochondria or cardiac-related gene expression changes focusing on reprogramming day 4. In addition, we have confirmed that CMs still showed normal cytoplasmic α MHC expression after doxycycline-induced reprogramming for 2 days as shown in the figure below.

Figure. Co-staining of H3P and α MHC to confirm that CMs maintain their identity after reprogramming induction for 2 days.

4. More quantitative methods (such as FACS analysis) should be used to quantify the percentage of proliferating cells.

We thank reviewer for this suggestion. We used transgenic adult M/Z mice which contain tamoxifen-induced α MHC promoter-driven EGFP to label adult CMs in green. After injection of adeno-Ctrl or adeno-FIJs directly into the adult hearts of M/Z mice and BrdU labeling for 4 days, the adult CMs were isolated at 12 days post-infection and the BrdU⁺/EGFP⁺ population % was measured by flow cytometry. The results are shown in Figure 4F that BrdU⁺/EGFP⁺ population % was 5-times more in FIJs-treated CMs than in control group.

Results

Page: 9

In order to further confirm that proliferated population did indeed originate from CMs, double transgenic mice myh6-mER-Cre-mER/ZEG were generated with tamoxifen-induced α MHC promoter-driven EGFP to label adult CMs in green. After injection of adeno-Ctrl or adeno-FIJs directly into the adult hearts of M/Z mice and BrdU labeling for 4 days (Fig 4F), the adult CMs were isolated 12 days post-infection and the BrdU⁺/EGFP⁺ population was 5 times higher in FIJs-treated adult CMs than control-treated CMs.

Figures with Figure Legends

Page: 31

Figure 4. The defined gene cocktail induces CM proliferation in neonatal and adult mice. F Quantification of BrdU⁺/EGFP⁺ population % in isolated CMs from adeno-Ctrl or adeno-FIJs injected M/Z mice.

5. Figure5c. The authors should demonstrate that the proliferating CM are the one overexpressing FIJs.

We thank the reviewer for the suggestion. The adenovirus that we used in this study carries EGFP and the infection was transient for 7 days. Therefore, we confirmed that EGFP expression in the heart tissues was localized to the area surrounding the adenovirus injection site, and cardiomyocyte proliferation was also confirmed as increased by co-staining of proliferative markers with EGFP or cTnI at the same day after MI (Fig 5). We have also confirmed that EGFP expression in adenoviral infected tissues was co-localized with FoxM1 or ID1 expression (Appendix Fig S6).

Figures with Figure Legends

Page: 32

Figure 5. Enhancement of CM proliferation by delivery of a gene cocktail after MI.

C Immunofluorescence staining of BrdU or H3P and cTnI or EGFP and quantification in the heart tissues after MI. The arrow head indicates cTnI⁺/BrdU⁺, cTnI⁺/H3P⁺ or EGFP⁺/BrdU⁺, EGFP⁺/H3P⁺ proliferated CMs.

Materials and Methods

Page: 19

Efficiency of Recombinant Adenoviral Infection *in vitro* and *in vivo*

The infection efficiency was more than 90% and the adenovirus-infected CMs with EGFP fluorescence were confirmed to express both FoxM1 and ID1 (Appendix Fig S6A). Overexpression of each specific gene was confirmed to be greater than 100-fold. For *in vivo* infection, the exogenous expression of FoxM1 or Id1 could be enhanced more than 2 times that of the endogenous expression after MI, and Jnk3 had 3 times lower exogenous expression than endogenous expression four days after adenoviral infection (Fig 5B). At the same time, EGFP expression in the heart tissues was confirmed localized to the area surrounding the adenovirus injection site near the injury site and co-localized with FoxM1 or ID1 expression (Appendix Fig S6B). However, after 7 days the induction vanished and FIJ expression was at the same level in the adeno-Ctrl or adeno-FIJs treated heart tissues (Fig 5B).

Supplemental Figures with Figure Legends

Page: 19

Appendix Fig S6. Adenoviral infection efficiency of CMs after adeno-FIJs treatment.

A EGFP expression combined with immunofluorescence staining of FoxM1 and Id1 in CMs after adeno-FIJs treatment *in vitro*.

B Immunofluorescence staining of EGFP and FoxM1 or Id1 in CMs after adeno-FIJs treatment *in vivo*. The scale bar represents 50 μ m.

6. Figure 5d. The authors should use cardiac MRI or cardiac catheterization to measure EF. Echocardiography is not a quantitative way for measuring EF. What about FS? The improvement of cardiac function after FIJs administration seems very mild. The authors should rephrase their statement in Page 2 line 19.

We thank the reviewer for this comment. We included the FS data in Figure 5D and showed the same improvement of heart function under FIJs treatment after injury.

Furthermore, we have repeated the same experiments and measured heart function by direct LV catheterization. The results, shown in Fig 5E, demonstrate that heart function was significantly improved by FIJs treatment following MI. These improvements were in line with those that we previously demonstrated by echocardiography, shown in Fig 5D.

Results

Page: 10

Delivery of Gene Cocktail FIJs Enhances Cardiomyocyte Proliferation for Improved Heart Function after Myocardial Infarction

At the same time, cardiac performance was also measured by catheterization, and the results showed a significantly higher dP/dt value in FIJs-treated mice under baseline condition. Higher ESPVR and PRSW and lower Tau value were confirmed in FIJs-treated mice under inferior vena cava occlusions, representing overall improved cardiac function 21 days after MI injury (Fig 5E).

Figures with Figure Legends

Page: 32

Figure 5. Enhancement of CM proliferation by delivery of a gene cocktail after MI.

D Ejection fraction and fraction shortening percentage in Ctrl or FIJs-treated mice 21 days post-MI.

E Catheterization in Ctrl or FIJs-treated mice 21 days post-MI under baseline or inferior vena cava occlusion condition

7. FoxM1 and Id1 are well known oncogenes, and Jnk3 is a tumor suppressor. The oncogenic activity of this cocktail may compromise its clinical potential. The authors should discuss on this possibility.

The reviewer is totally correct about this point and we have included some discussion of this concern in the Discussion section of the manuscript. FIJs treatment should be transient to induce CM proliferation for heart repair after injury, and so the time point, duration and delivery method of FIJs treatment would need to be carefully determined to evaluate its clinical application for heart regeneration after injury.

Discussion

Page: 13

Compared to the previously mentioned single or double treatments, our gene cocktail changed expression of 3 specific genes to induce higher CM proliferation (7 times) which led to efficient improvement of heart function after injury. It would be the next step to determine the concentration and ratio of FIJs cocktail to reach the optimized utilization to increase CM proliferation for heart regeneration after injury. We do not rule out the possibility that other mechanisms regulate this treatment in CMs, and the oncogenic activity of FIJs cocktail may be the limitation to compromise its clinical application. Therefore, the time point, duration, and long-term effects of FIJs treatment should be determined to carefully evaluate its clinical application in the future.

8. - Supp Fig3, the down-regulation of Id1 and the up-regulation of Jnk3 expression do not coincide with the loss of CM proliferation.

Id1 has been declared as essential for preventing senescence in human fetal cardiomyocytes, implying its high expression in fetal cardiomyocytes with proliferative potential (Ball and Levine 2005). In addition, in cancer studies, Id1 has been shown to be essential for completing the mitosis process (Wang, Di et al. 2008) – the failure of which is reported to lead to the binucleation of cardiomyocytes, instead of proliferation (Li, Wang et al. 1997, Li, Wang et al. 1997). In our study, we also confirmed that Id1 single treatment enhanced CDK1 and some mitotic gene expression compared to the FoxM1 or shJnk3 single treatment (Figure 6), and the combined treatment showed the best support for complete mitosis of cardiomyocytes, compared to the single treatments alone. Therefore, even Id1 does not totally correlate with neonatal cardiomyocyte proliferation, its addition to the FIJs cocktail certainly supports the completion of cytokinesis for cardiomyocyte proliferation.

The Jnk family is implicated as important in the heart field, and Jnk3, specifically expressed in heart and brain, showed a supportive role for proliferation in brain cells (Mielke 2008). Hence, Jnk3 may play the same supportive role for cardiomyocyte proliferation, and our data does support this hypothesis to enhance cardiomyocyte proliferation through decreasing p27, as shown in Figures 2 and 6. Therefore, the combined cocktail FIJs may relate to the natural process of heart biology and shows therapeutic potential for heart diseases.

We agree that the patterns of expression during fetal development do not coincide perfectly with CM proliferation, which is why this information was provided as supporting supplementary information to provide context.

Reviewer #3

1. The *in vitro* study for cardiomyocyte proliferation was conducted in day 1-3 neonatal cardiomyocytes which already have a relatively high baseline of cell proliferation as indicated in Figure 3D and E. The control has least 3-5% proliferating cells, which goes up to 10-15% with the FIJs cocktail. It seems preferable to use more mature cardiomyocytes to conduct such experiments, e.g. day 7-10 cardiomyocyte.

We thank the reviewer for these comments and we completely agree that using neonatal cardiomyocytes is a limitation of this study due to their relatively high baseline of proliferation, and relative ease of proliferation enhancement. We have made several attempts at isolating adult cardiomyocytes for use in these experiments, however we were unable to culture primary adult or day 7-10 cardiomyocytes for long enough *in vitro* to successfully infect with adenovirus and measure their proliferation.

Therefore, as a more direct alternative, we injected FIJs cocktail into adult hearts *in vivo*, and measured adult CM proliferation by tissue sections with immunofluorescence. The results showed 3-times more proliferation population % (H3P⁺/cTnI⁺) in FIJs-treated adult CMs compared to the control group (Fig 4E). Furthermore, we have performed the same experiments in adult M/Z mice with EGFP-labeled CMs, and measured the BrdU⁺/EGFP⁺ population % of isolated adult CMs by flow cytometry. As shown in the Figure 4F, we obtained the same results showing 5-times more BrdU⁺/EGFP⁺ population % in FIJs treated adult CMs compared to the control group.

Results

Page: 9

In order to further confirm that proliferated population did indeed originate from CMs, double transgenic mice *myh6-mER-Cre-mER/ZEG* were generated with tamoxifen-induced *aMHC* promoter-driven *EGFP* to label adult CMs in green. After injection of adeno-Ctrl or adeno-FIJs directly into the adult hearts of M/Z mice and BrdU labeling for 4 days (Fig 4F), the adult CMs were isolated 12 days post-infection and the $\text{BrdU}^+/\text{EGFP}^+$ population was 5 times higher in FIJs-treated adult CMs than control-treated CMs.

Figures with Figure Legends

Page: 31

Figure 4. The defined gene cocktail induces CM proliferation in neonatal and adult mice. F Quantification of $\text{BrdU}^+/\text{EGFP}^+$ population % in isolated CMs from adeno-Ctrl or adeno-FIJs injected M/Z mice.

2. The authors used end point staining for Ki67, phospho histone H3 and Aroua Kinase B as proof of cardiomyocyte cell division *in vitro*. Since each marker has its limitations on truly interpreting the presence of cell division, it would be useful if the authors showed time lapse video of the cell division and how many cells undergo cytokinesis and whether the cells survive or not after division and for how long.

We thank the reviewer for this helpful suggestion. In order to confirm that FIJs-induced CMs undergo the complete cell cycle, particularly cell division, we used time-lapse video to capture the cell division of CMs treated with FIJs, as shown in Appendix Fig S2A. Furthermore, we have calculated the total numbers of CMs in each whole well of 384-well plate after FIJs treatment in order to measure how many induced CMs undergo complete cytokinesis to increase cell numbers. As shown in the Appendix Fig S2B, the total numbers of FIJs-treated CMs were 5-times more than control group *in vitro*.

In terms of survival following FIJs treatment, CMs survived until at least day 3 where we measured their proliferation *in vitro*. More importantly, following FIJs injection *in vivo*, the survival of mice was not affected and FIJs treatment showed only positive effects on cardiac function. In addition, TUNEL assay was used to measure the percentage of apoptotic cells one day after myocardial infarction followed with adeno-Ctrl or adeno-FIJs injection (n = 4 animals per group), and the apoptotic cells were labeled as TUNEL-positive cells. The percentage of the TUNEL-positive cardiomyocytes did not show any significant difference between the control and FIJs-treated groups, as shown in Figure below.

Figure. TUNEL assay for measuring apoptotic CMs under adeno-Ctrl or adeno-FIJs treatment after MI.

Results

Page: 8

Determination of the gene cocktail with the potential of increasing cardiomyocyte proliferation in vitro and in vivo

Furthermore, time-lapse video showed that FIJs-treated CMs complete the whole cell cycle to produce 2 daughter cells, and total cell numbers were calculated to show more than 5-times increase in FIJs-treated CMs than control group (Appendix Fig S2A-B).

Supplementary Figures with Figure Legends

Page: 14

Figure S1. FIJs-treated CMs undergo complete cell cycle to increase total cell numbers

A The time-lapse microscopy of the complete cell cycle of isolated CMs

B Total numbers of isolated CMs after treating with adeno-Ctrl or adeno-FIJs

3. The *in vivo* study did not show convincing evidence that there is proliferation resulting in mature functional cardiomyocytes and there is a need to use lineage tracing mouse models to prove this finding. For example, mosaic analysis of dual marker (MADM) mice (Ali et al., PNAS. 2014;111(24):8850-5) or some such system would be much more rigorous, particularly given the notorious difficulty in ensuring that cell division of cardiomyocytes actually occurred *in vivo*.

We thank reviewer for the comments. To address this concern we have used transgenic adult M/Z mice, which contain tamoxifen-induced α MHC promoter-driven EGFP, to label adult CMs in green. After injection of adeno-Ctrl or adeno-FIJs directly into the adult hearts of M/Z mice and BrdU labeling for 4 days, the adult CMs were isolated 12 days post-infection and the BrdU⁺/EGFP⁺ population % was measured by flow cytometry. The results, as shown in Figure 4F, show that the BrdU⁺/EGFP⁺ population % was 5-times more in FIJs-treated hearts than in the control group.

Results

Page: 9

Determination of the gene cocktail with the potential of increasing cardiomyocyte proliferation in vitro and in vivo

In order to further confirm that proliferated population did indeed originate from CMs, double transgenic mice myh6-mER-Cre-mER/ZEG were generated with tamoxifen-induced α MHC promoter-driven EGFP to label adult CMs in green. After injection of adeno-Ctrl or adeno-FIJs directly into the adult hearts of M/Z mice and BrdU labeling for 4 days (Fig 4F), the adult CMs were isolated 12 days post-infection and the BrdU⁺/EGFP⁺ population was 5 times higher in FIJs-treated adult CMs than control-treated CMs.

Figures with Figure Legends

Page: 31

Figure 4. The defined gene cocktail induces CM proliferation in neonatal and adult mice.

F Quantification of BrdU⁺/EGFP⁺ population % in isolated CMs from adeno-Ctrl or adeno-FIJs injected M/Z mice.

4. The authors should show the efficiency of overexpressing Id1 and FoxM1 and knockdown of JNK3 *in vitro* and *in vivo* on protein level using western blot or immunofluorescence. It is mentioned in the methods section and qPCR data *in vivo* is shown in Figure 5B.

We thank the reviewer for the suggestion. We have confirmed that the expressional changes of FIJs induced by adenoviral infection were transient. For the *in vitro* experiments, cardiomyocytes showed EGFP until the day that we measured the proliferation assay and most adenovirus-infected cardiomyocytes with EGFP fluorescence were confirmed to express both FoxM1 and ID1 proteins (Appendix Fig S6A). For the *in vivo* experiments, we have also confirmed that EGFP expression in the heart tissues was localized to the area surrounding the adenovirus injection site, and cardiomyocyte proliferation was also confirmed by co-staining of proliferative markers with EGFP or cTnI at the same day after myocardial infarction (Fig 5). In addition, we have also confirmed that EGFP expression in adenoviral infected tissues was co-localized with FoxM1 or ID1 expression (Appendix Fig S6B). However, after 7 days, the induction vanished and FIJ expression returned to the same level as the adeno-Ctrl treated heart tissues.

Appendix Fig S6. Adenoviral infection efficiency of CMs after adeno-FIJs treatment.

A EGFP expression combined with immunofluorescence staining of FoxM1 and Id1 in CMs after adeno-FIJs treatment in vitro.

B Immunofluorescence staining of EGFP and FoxM1 or Id1 in CMs after adeno-FIJs treatment in vivo. The scale bar represents 50 μ m.

5. In figure 5D, it is surprising that the ejection fraction after 1 day went dramatically down and stayed at the same value after 3 weeks. Usually it takes 3-7 days for the scar to stabilize so the function stays partially preserved at day 1 and then there is typically progressive dysfunction after 3 weeks.

We thank the reviewer for raising this concern. The myocardial infarction surgery that we performed is a permanent ligation model, and so a massive amount of CMs undergo death immediately after MI resulting in the dramatic low ejection fraction one day post-MI, as shown in Fig 5D. Also, this surgery is a routine and reproducible technique in our lab (Hsieh, Segers et al. 2007, Lin, Luo et al. 2012).

6. The *in vivo* study in Figure 5 shows very subtle improvement in cardiac function from ~35% to ~40% (Figure 5D) and based on that subtle difference the authors claimed that there is very subtle induction in proliferation (3 to 8 H3P⁺ cells out of ~4000) which will potentially treat heart failure. The current data are not convincing and the authors should track the proliferative cells *in vivo* and determine what fate they take. Are these cells from cardiomyocytes that divided or are they from resident cardiac progenitors, if any, that divided?

We thank the reviewer for the suggestion. We used transgenic adult M/Z mice which contain tamoxifen-induced α MHC promoter-driven EGFP to label adult CMs in green. After injection of adeno-Ctrl or adeno-FIJs directly into the hearts of adult M/Z mice and BrdU labeling for 4 days, the adult CMs were isolated 12 days post-infection and the BrdU⁺/EGFP⁺ population % was

measured by flow cytometry. The results, shown in Fig 4F, demonstrate that BrdU⁺/EGFP⁺ population % was 5-times more in FIJs-treated hearts than in the control group. In addition, we also measured the BrdU⁺/EGFP⁻ population % which may be the new-born CM derived from the cells including cardiac progenitors of the hearts other than CMs by flow cytometry. As shown in the figure 4F, only a very low number of BrdU⁺/EGFP⁻ population were found in control or FIJs-treated groups. This indicates that most of the proliferated CMs are indeed derived from the resident CMs.

Results

Page: 9

Determination of the gene cocktail with the potential of increasing cardiomyocyte proliferation in vitro and in vivo

In order to further confirm that proliferated population did indeed originate from CMs, double transgenic mice myh6-mER-Cre-mER/ZEG were generated with tamoxifen-induced α MHC promoter-driven EGFP to label adult CMs in green. After injection of adeno-Ctrl or adeno-FIJs directly into the adult hearts of M/Z mice and BrdU labeling for 4 days (Fig 4F), the adult CMs were isolated 12 days post-injection and the BrdU⁺/EGFP⁺ population was 5 times higher in FIJs-treated adult CMs than control-treated CMs.

Figures with Figure Legends

Page: 31

Figure 4. The defined gene cocktail induces CM proliferation in neonatal and adult mice. F Quantification of BrdU⁺/EGFP⁺ population % in isolated CMs from adeno-Ctrl or adeno-FIJs injected M/Z mice.

7. The in vivo study was conducted for only 3 weeks and usually to obtain chronic heart failure 8-12 weeks are needed. 3 weeks is too short and is insufficient time to make conclusions.

We thank the reviewer for raising this concern. However, the rodent heart undergoes rapid tissue remodeling after MI. Around 7 days after MI, the infarct LV wall becomes thin, and the cardiac function decreases in a corresponding manner, reaching 20-30% decrease of EF% at 2-3 weeks. Thus, many groups, including us, use within 1 month after MI as the initial end-point (Hsieh, Davis et al. 2006, Lin, Luo et al. 2012, Chang, Yang et al. 2013, Chen, Wang et al. 2013).

In addition, as per EMBO MM current policy, we need to submit the revised manuscript within 3 months and are unable to carry out such a long-term experiment during this revision. However, we accept this limitation and we have written additional information into the Discussion section of the manuscript to address this issue.

Discussion

Page: 17

Compared to the previously mentioned single or double treatments, our gene cocktail changed expression of 3 specific genes to induce higher CM proliferation (7 times) which led to efficient improvement of heart function after injury. It would be the next step to determine the concentration and ratio of FIJs cocktail to reach the optimized utilization to increase CM proliferation for heart regeneration after injury. We do not rule out the possibility that other mechanisms regulate this treatment in CMs, and the oncogenic activity of FIJs cocktail may be the limitation to compromise its clinical application. Therefore, the time point, period, and long-term effects of FIJs treatment should be determined to carefully evaluate its clinical application in the future.

References

- Ball, A. J. and F. Levine (2005). "Telomere-independent cellular senescence in human fetal cardiomyocytes." *Aging Cell* 4(1): 21-30.
- Cao, N., Y. Huang, J. S. Zheng, C. I. Spencer, Y. Zhang, J. D. Fu, B. M. Nie, M. Xie, M. L. Zhang, H. X. Wang, T. H. Ma, T. Xu, G. L. Shi, D. Srivastava and S. Ding (2016). "Conversion of human fibroblasts into functional cardiomyocytes by small molecules." *Science* 352(6290): 1216-1220.
- Chang, M. Y., Y. J. Yang, C. H. Chang, A. C. L. Tang, W. Y. Liao, F. Y. Cheng, C. S. Yeh, J. J. Lai, P. S. Stayton and P. C. H. Hsieh (2013). "Functionalized nanoparticles provide early cardioprotection after acute myocardial infarction." *Journal of Controlled Release* 170(2): 287-294.
- Chen, C. H., S. S. Wang, E. I. H. Wei, T. Y. Chu and P. C. H. Hsieh (2013). "Hyaluronan Enhances Bone Marrow Cell Therapy for Myocardial Repair After Infarction." *Molecular Therapy* 21(3): 670-679.
- Hattori, F., H. Chen, H. Yamashita, S. Tohyama, Y. S. Satoh, S. Yuasa, W. Li, H. Yamakawa, T. Tanaka, T. Onitsuka, K. Shimoji, Y. Ohno, T. Egashira, R. Kaneda, M. Murata, K. Hidaka, T. Morisaki, E. Sasaki, T. Suzuki, M. Sano, S. Makino, S. Oikawa and K. Fukuda (2010). "Nongenetic method for purifying stem cell-derived cardiomyocytes." *Nat Methods* 7(1): 61-66.
- Hsieh, P. C. H., M. E. Davis, J. Gannon, C. MacGillivray and R. T. Lee (2006). "Controlled delivery of PDGF-BB for myocardial protection using injectable self-assembling peptide nanofibers." *Journal of Clinical Investigation* 116(1): 237-248.
- Hsieh, P. C. H., V. F. M. Segers, M. E. Davis, C. MacGillivray, J. Gannon, J. D. Molkentin, J. Robbins and R. T. Lee (2007). "Evidence from a genetic fate-mapping study that stem cells refresh adult mammalian cardiomyocytes after injury." *Nature Medicine* 13(8): 970-974.
- Li, F., X. Wang, P. C. Bunger and A. M. Gerdes (1997). "Formation of binucleated cardiac myocytes in rat heart: I. Role of actin-myosin contractile ring." *J Mol Cell Cardiol* 29(6): 1541-1551.
- Li, F., X. Wang and A. M. Gerdes (1997). "Formation of binucleated cardiac myocytes in rat heart: II. Cytoskeletal organisation." *J Mol Cell Cardiol* 29(6): 1553-1565.
- Lin, Y. D., C. Y. Luo, Y. N. Hu, M. L. Yeh, Y. C. Hsueh, M. Y. Chang, D. C. Tsai, J. N. Wang, M. J. Tang, E. I. H. Wei, M. L. Springer and P. C. H. Hsieh (2012). "Instructive Nanofiber Scaffolds with VEGF Create a Microenvironment for Arteriogenesis and Cardiac Repair." *Science Translational Medicine* 4(146).
- Mielke, K. (2008). "Growth-arrest-dependent expression and phosphorylation of p27kip at serine10 is mediated by the JNK pathway in C6 glioma cells." *Molecular and Cellular Neuroscience* 38(3): 301-311.
- Polizzotti, B. D., B. Ganapathy, S. Walsh, S. Choudhury, N. Ammanamanchi, D. G. Bennett, C. G. dos Remedios, B. J. Haubner, J. M. Penninger and B. Kuhn (2015). "Neuregulin stimulation of cardiomyocyte regeneration in mice and human myocardium reveals a therapeutic window." *Science Translational Medicine* 7(281).
- Rizzi, R., E. Di Pasquale, P. Portararo, R. Papait, P. Cattaneo, M. V. Latronico, C. Altomare, L. Sala, A. Zaza, E. Hirsch, L. Naldini, G. Condorelli and C. Bearzi (2012). "Post-natal cardiomyocytes can generate iPSC cells with an enhanced capacity toward cardiomyogenic re-differentiation." *Cell Death Differ* 19(7): 1162-1174.
- Wang, X., K. Di, X. Zhang, H. Y. Han, Y. C. Wong, S. C. Leung and M. T. Ling (2008). "Id-1 promotes chromosomal instability through modification of APC/C activity during mitosis in response to microtubule disruption." *Oncogene* 27(32): 4456-4466.
- Xu, H., B. A. Yi, H. Wu, C. Bock, H. Gu, K. O. Lui, J. H. Park, Y. Shao, A. K. Riley, I. J. Domian, E. Hu, R. Willette, J. Lepore, A. Meissner, Z. Wang and K. R. Chien (2012). "Highly efficient derivation of ventricular cardiomyocytes from induced pluripotent stem cells with a distinct epigenetic signature." *Cell Res* 22(1): 142-154.

2nd Editorial Decision

22 November 2016

Thank you for the submission of your revised manuscript to EMBO Molecular Medicine and for your patience during the review period. Unfortunately we did experience delay as one referee failed to provide her/his report; we have decided to make a decision based on the two reports received from two of the referees that were asked to re-assess it. As you will see the reviewers are now globally supportive and I am pleased to inform you that we will be able to accept your manuscript pending final editorial amendments.

Please submit your revised manuscript within two weeks. I look forward to seeing a revised form of your manuscript as soon as possible.

***** Reviewer's comments *****

Referee #1 (Remarks):

I have re-reviewed the current revised manuscript and find that they have adequately addressed my previous issues. I favour publication.

Referee #2 (Remarks):

The authors have addressed all of my questions. I recommend this paper for publication. Please let me know if you have any questions.

Corresponding Author Name: Patrick C.H. Hsieh

Journal Submitted to: EMBO Molecular medicine

Manuscript Number: EMM-2016-06558